# Z-flipon variants reveal the many roles of Z-DNA and Z-RNA in health and disease

Dmitry Umerenkov[1,*] , Alan Herbert[2,3,*]† , Dmitrii Konovalov[2] , Anna Danilova[2] , Nazar Beknazarov[2], Vladimir Kokh[1], Aleksandr Fedorov[2] , Maria Poptsova[2]

Identifying roles for Z-DNA remains challenging given their dynamic nature. Here, we perform genome-wide interrogation with the DNABERT transformer algorithm trained on experimentally identified Z-DNA forming sequences (Z-flipons). The algorithm yields large performance enhancements (F1 = 0.83) over existing approaches and implements computational mutagenesis to assess the effects of base substitution on Z-DNA formation. We show Z-flipons are enriched in promoters and telomeres, overlapping quantitative trait loci for RNA expression, RNA editing, splicing, and disease-associated variants. We cross-validate across a number of orthogonal databases and define BZ junction motifs. Surprisingly, many effects we delineate are likely mediated through Z-RNA formation. A shared Z-RNA motif is identified in SCARF2, SMAD1, and CACNA1 transcripts, whereas other motifs are present in noncoding RNAs. We provide evidence for a Z-RNA fold that promotes adaptive immunity through alternative splicing of KRAB domain zinc finger proteins. An analysis of OMIM and presumptive gnomAD loss-of-function datasets reveals an overlap of Z-flipons with disease-causing variants in 8.6% and 2.9% of Mendelian disease genes, respectively, greatly extending the range of phenotypes mapped to Z-flipons.

## Introduction

The discovery of the Zα domain in the p150 isoform of the dsRNA-editing enzyme ADAR1 (encoded by ADAR), along with genetic studies in both humans (Herbert, 2020a) and mice (de Reuver et al, 2022; Hubbard et al, 2022; Jiao et al, 2022), has unambiguously confirmed a biological role for both Z-DNA and Z-RNA (collectively called ZNA) in the regulation of interferon responses, self/nonself transcript discrimination (Herbert, 2021b) and the necroptosis cell death pathways (Zhang et al, 2022). The covalent modifications of adenosine-to-inosine (A→I) RNA editing performed by ADAR1 and the MLKL phosphorylation activated by ZBP1 (ZNA-binding protein 1) enabled tracking of transient ZNA formation in cells.

Here, we use a genome-wide approach to discover additional phenotypes that are regulated by Z-flipons, sequences that can form ZNAs under physiological conditions. Our approach is computational and employs a novel and highly efficient algorithm for predicting Z-flipons based on experimental data. We leverage the large number of orthogonal datasets from the human genome and ENCODE projects to evaluate the validity of many hypotheses and presented here are those that are not falsified by existing experimental evidence.

We started with a pretrained DNABERT model (Ji et al, 2021) and fine-tuned it with validated Z-flipons from human genome-wide experimental studies (Fig 1). The resulting Z-DNABERT significantly outperformed previous approaches such as DEEPZ (Beknazarov et al, 2020) that are based on convolutional and recurrent neural networks, with a recall of 0.89, precision of 0.78, and ROC AUC of 0.99 (Table 1). The algorithm generates easily interpretable attention maps of Z-prone sequences at nucleotide resolution (Figs 1 and S1).

Our approach starts with the experimental permanganate/S1 nuclease dataset (KEx) from Kouzine et al that is based on mapping unpaired thymines present in the two BZ junctions formed with B-DNA at either end of a Z-DNA helix (Kouzine et al, 2017). After training of Z-DNABERT with this dataset, we compared the predictions with those from orthogonal approaches based on Z-HUNT3 (Ho, 2009) and kethoxal-assisted sequencing (K-seq). Z-HUNT3 is based on in vitro measurements capturing the energetic cost of flipping a base pair from B-DNA to Z-DNA, using a fixed energy cost for the formation of two BZ junctions. It estimates the propensity of a sequence to form Z-DNA in supercoiled DNA. K-seq uses chemical modification of unpaired guanosine bases with azide-tagged kethoxal performed with intact cells. The reaction detects regions of single-stranded DNA (ssDNA) arising from active transcription and R-loop formation (Weng et al, 2020; Wu et al, 2020). Unlike $KMnO_4$ that detects the unpaired base at a BZ junction, K-seq captures the opening of a Z-forming sequence as it flips from one

[1]Sber Artificial Intelligence Lab, Moscow, Russia   [2]Laboratory of Bioinformatics, Faculty of Computer Science, HSE University, Moscow, Russia   [3]InsideOutBio, Charlestown, MA, USA

Correspondence: alan.herbert@insideoutbio.com; mpoptsova@hse.edu
*Dmitry Umerenkov and Alan Herbert are co-first authors
†Alan Herbert is communicating author

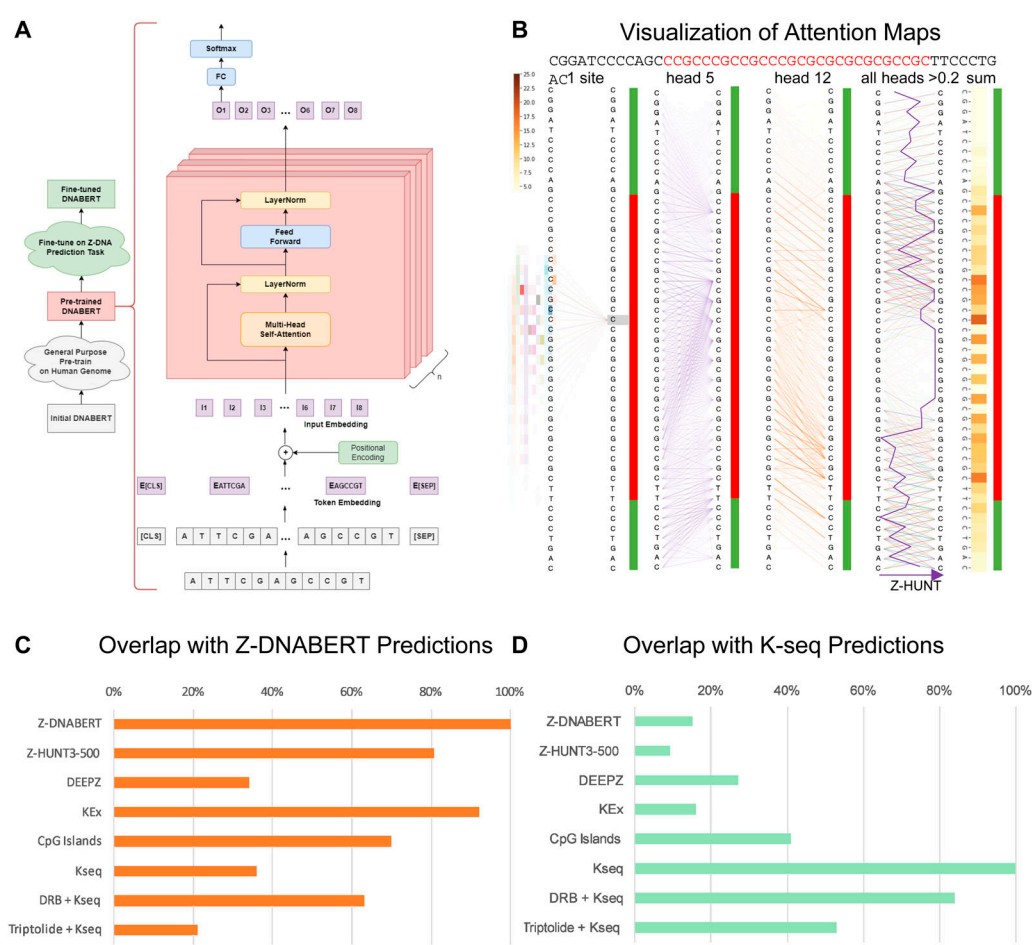

**Figure 1. Generation of whole-genome Z-flipon maps with the Z-DNABERT model.**
**(A)** Architecture of Z-DNABERT showing fine-tuning of DNABERT on experimental Z-DNA datasets. **(B)** Interpretation of Z-DNABERT model. Visualization of attention scores for the sequence shown at the top of the panel that has the experimentally validated Z-DNA region colored red. From left to right: attention map for a single nucleotide; attention map from head 5; attention map from head 12; attention map output that combines all layers with the threshold >0.2; a line showing Z-HUNT3 scores across the sequence; heatmap summarizing Z-DNA propensity. **(C)** Overlap of Z-DNABERT predictions with experimentally mapped alternative conformations using potassium permanganate ($KMnO_4$, KEx) and ketoxal (Kseq) mapping, along with predictions made using DEEPZ and Z-HUNT3. Triptolide inhibits formation of the RNA polymerase preinitiation transcription complex, whereas DRB (5,6-Dichloro-1-b-D-ribofuranosylbenzimidazole) inhibits the release of RNA polymerase paused downstream of the transcription start site. **(D)** Overlap of different genome-wide methods for Z-DNA mapping with K-seq data.

**Table 1. Comparison of Z-DNABERT with other Z-DNA prediction ML models.**

| Validations on the test set: | | Precision | Recall | F1 | ROC AUC |
|---|---|---|---|---|---|
| Human Shin et al *Zaa ChIP-seq on HeLa cells* | DEEPZ | 0.59 | 0.56 | 0.57 | 0.94 |
| | Z-DNABERT | 0.68 | 0.43 | 0.53 | 0.95 |
| | CatBoost | 0,70 | 0.27 | 0.39 | 0.92 |
| Human Kouzine et al (KEx) | DEEPZ[a] | 0.01 | 0.30 | 0.02 | 0.89 |
| | Z-DNABERT | 0.78 | 0.89 | 0.83 | 0.99 |
| | CatBoost | 0.01 | 0.17 | 0.03 | 0.98 |

[a]DeepZ that was trained on Shin et al data.

conformation to another. The half-life for the BZ transition in vitro under physiological levels of supercoiling is estimated to be 100 ms. Consequently, the reaction of kethoxal with DNA is measured over 5–10 min compared with the 70 s used for $KMnO_4$ modification and does not provide specific information about junctions (Jovin et al, 1987). Here, we use the overlap between each of these different predictive and experimental approaches to map Z-DNA formation to specific genomic loci. We confirm the enrichment of Z-flipons in

promoters and identify Z-DNA–prone repeat families. We also identify motifs in BZ junctions that are identified by Z-DNABERT, confirming that they form preferentially by an eversion of an adenosine from the helix (Kim et al, 2018).

We then used the large number of datasets available to map DNA variants affecting Z-DNA formation to the phenotype, enabling us to perform a deep analysis of how flipons encode genetic information. We focused on Z-DNA regions experimentally verified by KEx, examining genomic variants previously identified by Genome-Wide Association Studies (GWAS) or found by disease-directed approaches. We performed computational mutagenesis with Z-DNABERT to test directly whether SNP alleles affected Z-DNA formation and then used haplotype analysis to map flipon alleles to trait values. We also assessed the role of Z-flipons in mendelian disease. Our findings expand the range of phenotypes attributable to Z-flipons beyond the human mendelian type I interferonopathies caused by loss of function (LOF) ADAR1 p150 variants (Herbert, 2020a). We reveal a role for Z-DNA consistent with a role in resetting chromatin structure and the potential for involvement of Z-RNA in such processes.

# Results

## Developing generalizable deep learning model for Z-DNA prediction

Currently, there are two human experimental datasets available that provide information on Z-DNA formation within human cells: the Shin et al ChIP-seq (chromatin immunoprecipitation followed by DNA sequencing of fragments) experiments with a resolution of 100–150 bp (Shin et al, 2016) and the experimentally based dataset from Kouzine et al (KEx) (Kouzine et al, 2017). Kouzine et al determined Z-formation by the overlap of unpaired thymines detected using permanganate/S1 nuclease sequencing (ssDNA-seq) with Z-DNA–forming sequences predicted by Z-HUNT3. The thymines subject to modification were used to define the two BZ junctions where B-DNA transition to Z-DNA. The approach employed a number of statistical corrections to identify ssDNA-seq signals solely due to RNA polymerase 2 transcription or from other sequence variations (Kouzine et al, 2017). The final set (KEx) with all non-B-DNA (NoB) structures annotated is referred to by the authors as "ssDNA + SMnB." Both Shin et al and Kouzine et al approaches were performed in intact cells and differed from an earlier approach where Zα was diffused into detergent permeabilized cells and then cross-linked to DNA using formaldehyde over a number of hours (Li et al, 2009).

For the deep learning model, we chose DNABERT pretrained with 6-mers representation. The approach is based on the Bidirectional Encoder Representations from Transformers (BERT) algorithm (Ji et al, 2021). We then trained the model further using the experimental datasets to create Z-DNABERT (Fig 1A, see the Materials and Methods section and Supplemental Data 1). We compared the performance of Z-DNABERT with two other machine learning methods: DEEPZ (Beknazarov et al, 2020) and Gradient Boosting (CatBoost realization) (Dorogush et al, 2018 Preprint). The latter approach also learns from k-mers representation (Table 1).

Z-DNABERT showed high performance on F1 (0.83) and ROC AUC (0.99) when tuned with the KEx set, far outperforming DEEPZ (0.02 and 0.89, respectively). Whereas Z-DNABERT is based on nucleotide-resolution sequence data, DEEPZ analyzes DNA fragments identified by pull-down of epigenetically modified histones or after proteins are chemically cross-linked to DNA. Part of the reason for the enhancement of Z-DNABERT over DEEPZ is shown by the Shin et al analysis. With DEEPZ, poor ZNA-forming sequences such as AAAAAA are enriched because of bystander effects with their effects over estimated due to the small number of 100–150 bp fragments analyzed (Table 2).

Z-DNABERT outputs attention maps that are easily visualized (Figs 1B and S1) using pixel intensity to represent the importance of a particular residue in promoting Z-DNA formation. In this way, one can analyze the output summarized for all self-attention heads or for a particular head. Unlike the black box results from neural nets, the zebra-stripe patterns produced are easily interpretable: they show the propensity of alternating purine/pyrimidine dinucleotide repeats to form Z-DNA. The dark stripes correspond to purine bases that flip from the *anti* to the *syn* conformation as the transition from the right-handed to the left-handed helix occurs. The preference for guanosine over adenosine and cytosines over thymidine reflects

**Table 2. The top 21 6-mers: Z-DNABERT attention rank versus the 6-mer frequency rank in the experimental datasets tested for tuning the model.**

| Attention rank | hg38 Kouzine et al | | hg38 Shin et al | |
|---|---|---|---|---|
| | 6-mer | Frequency | 6-mer | Frequency |
| 1 | GCGCGC | 1 | TGTGTG | 1 |
| 2 | GTGTGT | 5 | GTGTGT | 2 |
| 3 | CGCGCG | 2 | CGCGCG | 4 |
| 4 | ACACAC | 6 | GCGCGC | 3 |
| 5 | TGTGTG | 3 | CACACA | 5 |
| 6 | GCGCGG | 7 | ACACAC | 6 |
| 7 | CACACA | 4 | GGGGAA | 40 |
| 8 | CCGCGC | 10 | AAAAAA | 17 |
| 9 | GGGCGC | 11 | CAGGGA | 43 |
| 10 | GCGCCC | 12 | GTGCGC | 11 |
| 11 | GTGCGC | 17 | TGGGGA | 331 |
| 12 | GGCGCG | 9 | GGGGGA | 39 |
| 13 | GTGTGC | 14 | GCTGGG | 9 |
| 14 | GCGCAC | 19 | GTGTGC | 7 |
| 15 | GCACAC | 15 | TGCGCG | 8 |
| 16 | GCCCGC | 20 | TGCATG | 21 |
| 17 | GCGGGC | 16 | GGGAAG | 33 |
| 18 | CGCGCC | 8 | AGGGAG | 429 |
| 19 | GCGTGC | 25 | GGGAGC | 458 |
| 20 | GCACGC | 26 | AGAAAG | 38 |
| 21 | CCCGCG | 18 | GGGAAA | 80 |

The model based on the experimental Kouzine et al data was used in the paper rather than the much smaller 150 bp resolution ChIP-seq data of Shin et al.

**Table 3. Z-DNABERT cross-species predictions.**

| Trained | Predict | Precision | Recall | F1 | ROC AUC |
|---------|---------|-----------|--------|-----|---------|
| Human Kouzine et al | hg Kouzine et al | 0.78 | 0.89 | 0.83 | 1.00 |
| Mouse Kouzine et al | hg Kouzine et al | 0.70 | 0.87 | 0.77 | 1.00 |

F1 = harmonic mean of precision and recall.

**Table 4. Z-DNABERT detection of repeats.**

| Rank | Repeat | Z-DNABERT overlap (n) | Percent | Total | Percent (expected) | Difference |
|------|--------|----------------------|---------|-------|--------------------|------------|
| 1 | L1 | 50,314 | 46.56% | 1,022,089 | 18.14% | 28.41% |
| 2 | Simple repeat | 31,303 | 28.97% | 717,938 | 12.74% | 16.22% |
| 3 | ERV1 | 6,942 | 6.42% | 185,322 | 3.29% | 3.13% |
| 4 | L2 | 3,028 | 2.80% | 482,724 | 8.57% | −5.77% |
| 5 | Alu | 2,899 | 2.68% | 1,269,382 | 22.53% | −19.85% |
| 6 | ERVL-MaLR | 2,171 | 2.01% | 364,558 | 6.47% | −4.46% |
| 7 | MIR | 2,109 | 1.95% | 612,281 | 10.87% | −8.92% |
| 8 | hAT-Charlie | 1,721 | 1.59% | 268,836 | 4.77% | −3.18% |
| 9 | ERVL | 1,720 | 1.59% | 170,629 | 3.03% | −1.44% |
| 10 | Low complexity | 1,604 | 1.48% | 105,114 | 1.87% | −0.38% |
| 11 | SVA | 1,604 | 1.48% | 5,913 | 0.10% | 1.38% |
| 12 | TcMar-Tigger | 606 | 0.56% | 122,222 | 2.17% | −1.61% |
| 13 | hAT-Tip100 | 555 | 0.51% | 47,409 | 0.84% | −0.33% |
| 14 | ERVK | 280 | 0.26% | 11,764 | 0.21% | 0.05% |
| 15 | Satellite | 246 | 0.23% | 5,394 | 0.10% | 0.13% |
| 16 | Centromere | 235 | 0.22% | 2,984 | 0.05% | 0.16% |
| 17 | CR1 | 152 | 0.14% | 69,112 | 1.23% | −1.09% |
| 18 | Gypsy | 74 | 0.07% | 17,283 | 0.31% | −0.24% |
| 19 | hAT-Blackjack | 55 | 0.05% | 19,950 | 0.35% | −0.30% |
| 20 | RTE-BovB | 43 | 0.04% | 9,247 | 0.16% | −0.12% |
| | Sum | 107,661 | 99.62% | 5,510,151 | 97.81% | |

the experimentally determined in vitro energetics that the Z-HUNT3 program uses to score Z-prone sequences (Ho, 2009). Compared with the Z-HUNT3 output ("all-heads" column Fig 1B), attention maps provide extra information on the sequence dependence of BZ junctions rather than assigning them a fixed energy cost. These additional details likely account for the slight differences in predicted ranking of Z-prone motifs compared with the experimental Z-DNA input data (Table 2). The Z-DNABERT model trained on human data also performed well in predicting Z-prone sequences from the mouse genome (Table 3). Z-DNABERT can further predict the effect on Z-DNA formation of substituting any nucleotide in a sequence with another.

## Whole-genome prediction of Z-flipons

With Z-DNABERT trained on the KEx experimental data (41,324 regions with total length of 773,788 bp), we generated genome-wide whole genome maps of Z-DNA–prone regions (Table S1), which resulted in 290,071 segments covering 3,167,809 bp (0.16%) of the hg38 genome build. The genomic coverage of predicted Z-flipons was much more extensive than that for KEx (Table S2). We observed many Z-DNABERT hits in repeat sequences, enabling us to analyze the performance of the model further (Tables 4 and S3). A major finding was the enrichment of predicted Z-DNA in LINEs (long interspersed nucleotide repeat elements), especially of the L1 family, and in simple repeats. Interestingly, the Alu SINEs (short interspersed nucleotide repeat elements) were under-represented by the Z-DNABERT algorithm, even though formation of Z-RNA by these elements by ADAR1 enables the negative regulation of interferon responses (Herbert, 2021b). However, these SINEs were filtered out by Kouzine et al and so they were not present in the Z-DNABERT training set (Kouzine et al, 2017). We further compared Z-HUNT3 and Z-DNABERT approaches by comparing the Z-DNA scores assigned to different simple repeat families (Tables 5 and S3). The ranking of and correlation between the top scores was high, even though Z-DNABERT was trained only on the KEx data that did not detect most of the predicted Z-HUNT3 loci. Both approaches scored even numbered repeats higher than odd-numbered repeats, consistent

**Table 5. Z-DNABERT scores for simple repeats.**

| Simple repeats | Repeat type | Bases in repeats | Mean base score | Max base score |
|---|---|---|---|---|
| 1 | (CG)n | 8,686 | 1.84 | 4.31 |
| 2 | (CACG)n | 26,369 | 1.03 | 4.40 |
| 3 | (CGGG)n | 2,141 | 1.03 | 4.37 |
| 4 | (CGTG)n | 30,000 | 1.00 | 4.38 |
| 5 | (CCCG)n | 1,988 | 0.86 | 4.34 |
| 6 | (CACGA)n | 390 | 0.47 | 4.11 |
| 7 | (CCGCG)n | 8,280 | 0.45 | 4.33 |
| 8 | (CGCGG)n | 9,225 | 0.42 | 4.42 |
| 9 | (CGAG)n | 2,205 | 0.27 | 4.18 |
| 10 | (GCGTG)n | 3,008 | 0.26 | 4.13 |
| 11 | (CAGCG)n | 690 | 0.25 | 2.81 |
| 12 | (CCTCG)n | 2,196 | 0.25 | 4.17 |
| 13 | (CATG)n | 40,401 | 0.24 | 4.20 |
| 14 | (CA)n | 3,020,329 | 0.24 | 4.41 |
| 15 | (TG)n | 3,010,676 | 0.23 | 4.40 |
| 16 | (CCCCCG)n | 6,484 | 0.20 | 4.23 |
| 17 | (CGGGGG)n | 6,746 | 0.17 | 4.21 |
| 18 | (CTCG)n | 1,741 | 0.16 | 3.97 |
| 19 | (CGGGG)n | 84,738 | 0.14 | 4.45 |
| 20 | (CCCCG)n | 72,871 | 0.14 | 4.40 |

with the dinucleotide *anti-syn* motif found in Z-DNA. One difference was the higher ranking of d(CGGG)n (and its complement d(CCCG)n) by Z-DNABERT than Z-HUNT3. The mapping of this repeat depends on the BZ junction as this sequence does not contain thymines sensitive to KMnO4 modification. The out-of-alternation purine–pyrimidine on this repeat confers only a small penalty compared with the 5 kcal/mol/dinucleotide per BZ junction, with the cost flipping d(CG) = 0.6 kcal/mol/dinucleotide, d(CA) = 1.34 kcal/mol/dinucleotide and d(GG) 2.4 = kcal/mol/dinucleotide) (Ho et al, 1986). Although these repeats could form G4-quadruplexes, the energetic cost would be higher as this transition requires the creation of four unpaired junction regions (2 at each end and 2 loops between strands) when the structure is formed by pairing two stem-loops, even more extensive unwinding is involved if G4-quadruplex formation depends on the intramolecular folding of four d(CGGG) repeat elements.

## Cross validation of Z-DNABERT with other Z-DNA detection methods

Z-DNABERT and K-seq genome-wide results were compared with Z-DNA maps generated by other methods (Fig 1C and D). In both cases, we evaluated the overlap with Z-HUNT3 (for scores >500, a threshold based on the results shown in Fig S2), DEEPZ (using the previously published threshold [Beknazarov et al, 2020]), the overlap with CpG islands and the effects of small chemical inhibitors that affect RNA polymerase initiation and processivity. The overlap of Z-DNABERT, KEx, and Z-HUNT3 exceeds 80%

genome-wide. With K-seq, the Z-DNABERT overlap is increased from around 36% to over 60% by DRB (5,6-Dichloro-1-b-D-ribofuranosylbenzimidazole), a compound inhibiting the release of an RNA polymerase paused near the transcription start site (TSS). In contrast, the Z-DNABERT overlap is diminished to around 20% by triptolide, an inhibitor of preinitiation transcription complex formation. The Z-DNABERT overlap with DEEPZ was around 35%, reflecting the limitations of this approach discussed above. For K-seq, the overlaps with Z-HUNT3, DEEPZ, and CpG islands was lower than for Z-DNABERT (Fig 1D). This result is expected as K-seq detects NoB conformations other than Z-DNA as well as transcription bubbles within gene bodies. Also expected is the increase in the promoter K-seq signal with DRB and the decrease with triptolide.

A genome browser view provides further insights as to how the mapping approaches differ from each other (Figs 2 and S3). In addition to tracks for each set of results, the image shows the localization of the negative elongation factor (NELF) components A and CD that cause pausing of RNA polymerase II just downstream of the TSS. Also displayed are the binding sites for AGO1 and AGO2, proteins guided by microRNA seed sequence matches with proximal promoter nucleotides (Herbert et al, 2023). All the sequencing methods reveal an increase in unpaired bases at promoters. The KEx approach adjusts for ssDNA formed in the absence of NoB structures by both probabilistic approaches based on randomizing counts in a region and by calculating expected counts after excluding SINE repeat sequences from their analysis. They also used thresholds to identify regions where the ssDNA-seq counts are

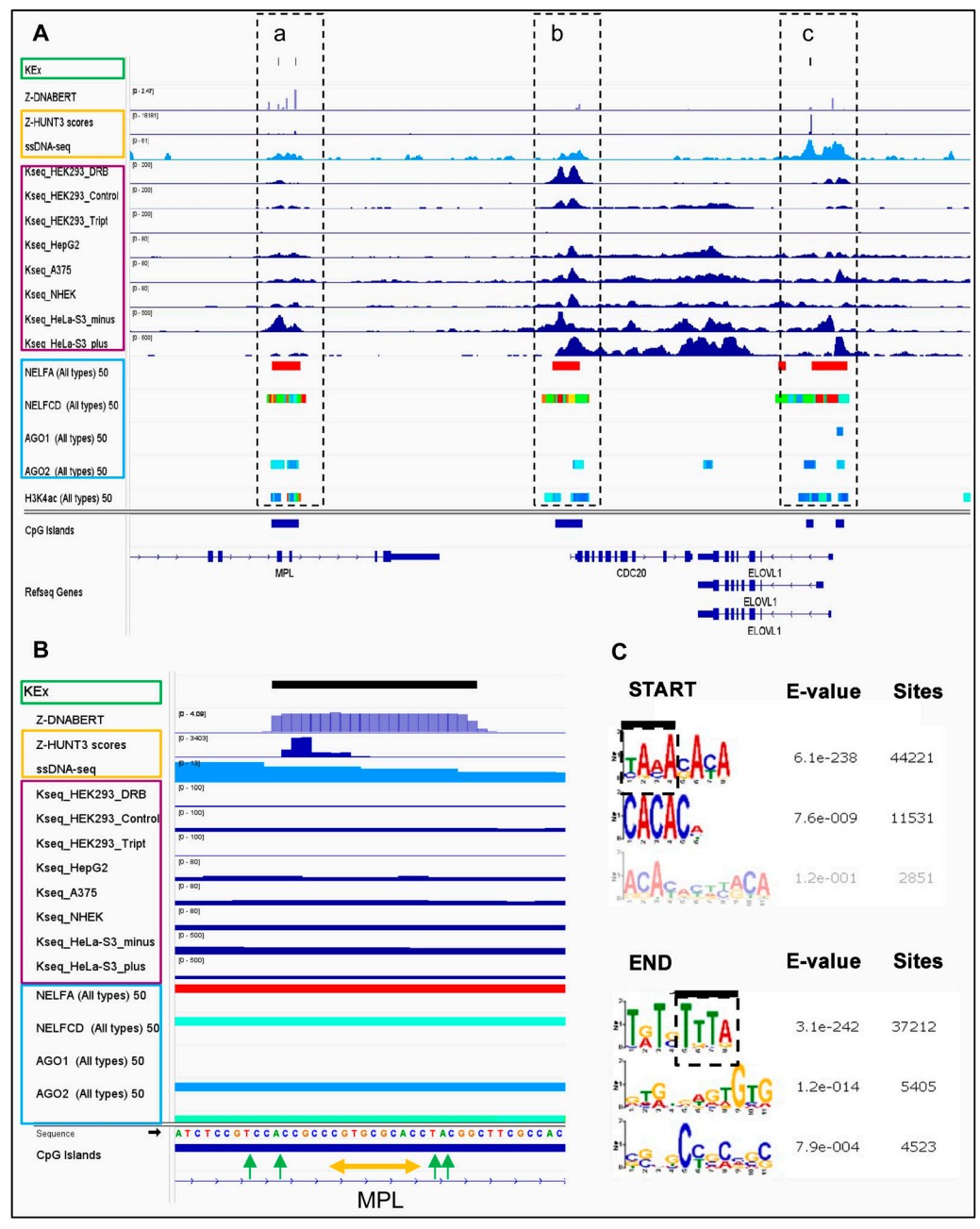

**Figure 2. Overlap of methods for detecting Z-flipons.**
**(A)** The different approaches show different patterns of overlap comparing Z-DNABERT with KEx (mapping of non-B-DNA structures, green boxes), ssDNA-seq (KMnO4 mapping of unpaired thymines), Z-HUNT3, K-seq (purple boxes) and protein-binding sites for negative elongation factors (NELF) of transcription and argonaute (AGO) proteins (blue boxes). The overlap of ssDNA-seq and Z-HUNT3 predictions (shown within the orange box) was used to map KEx Z-DNA structures (Kouzine et al, 2017). The published K-seq results are from different human cell lines (human embryonic kidney cells [HEK293], the hepatoma HepG2 line, the A375 melanoma line, normal human epidermal keratinocytes, and from the human cervical cancer HeLa line). Cells were treated with DRB (5,6-Dichloro-1-b-D-ribofuranosylbenzimidazole) or triptolide (Tript) inhibitors to examine the effects of RNA polymerase on K-seq maps. Strand-specific sequencing (plus or minus) was performed on normal human epidermal keratinocytes cells. Panel (a) shows that the concordance varies by cell line but that each method can detect a pattern of Z-DNA formation in the same promoter. Panel b reveals the complexity of K-seq as it can detect alternative DNA conformations other than produced by Z-DNA as also shown by the signals present in the body of a gene between panels (b, c). **(B)** Mapping of the overlap in panel a to nucleotide signal for Z-DNA detected by all four methods. Z-HUNT3 identifies the start of the Z-prone sequence that is anchored by the CGTGCGCA core (above the orange double-headed arrow) and can extend either side even though the alternating purine/pyrimidine motif is not conserved. Potential thymines modified by KMNO4 at the BZ junctions in KEx are indicated by green arrows. Both KMnO4 and Z-DNABERT identify the Z-DNA-forming region, whereas the resolution of Kseq is lower because of the detection of unpaired bases at other sites of Z-DNA formation. The region shown is from hg19 chr1: 43,809,400–43,838,608. **(C)** The start and end of Z-DNABERT predictions were analyzed by MEME to find BZ junction motifs, with the adenosines in this region highlighted with a thick dark line and the dashed box.

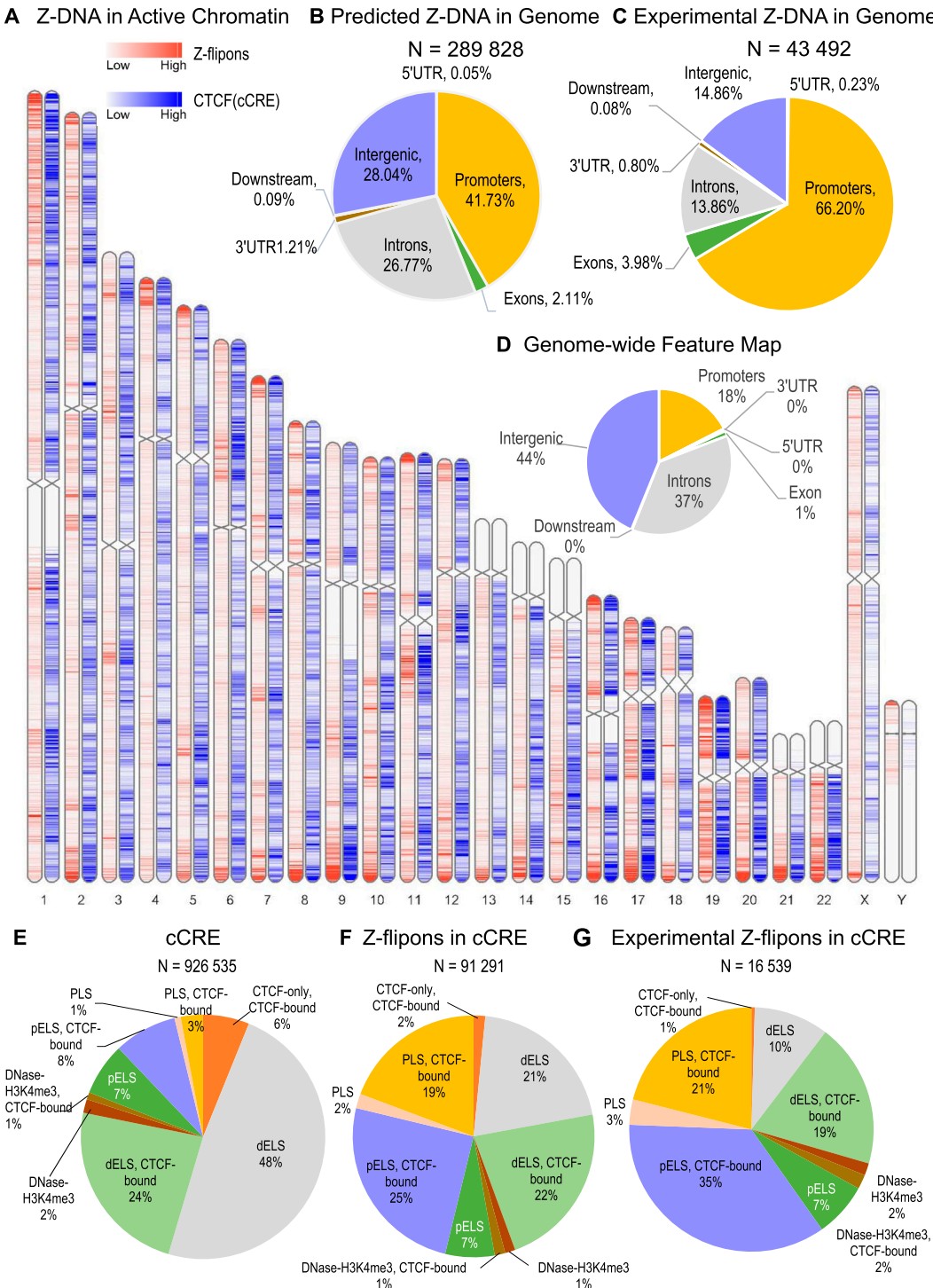

**Figure 3. Z-flipon overlaps with genomic features and conserved ENCODE cis Regulatory Elements (cCRE).**
**(A)** A whole-genome map with predicted Z-flipons compared with a map of CTCF protein-binding sites that are present in candidates in cCRE. **(B)** Genomic features of the predicted Z-flipons. **(C)** Genomic features of the experimental Z-flipons. **(D)** Background distribution of genomic features. **(E)** Genome-wide distribution of cCRE. **(F)** Z-DNABERT-predicted Z-flipon overlaps with cCRE. **(G)** KMnO₄-mapped (KEx) Z-flipon overlaps with cCRE.

twofold higher than expected from RNA polymerase 2 transcription (Kouzine et al, 2017). The regions with excess ss-DNA hits were overlapped with predictions of Z-HUNT3 to define the regions of Z-DNA formation shown here in the KEx track. No filtering was performed for K-seq results with the strand-specific sequencing plus and minus tracks from HeLa cells revealing the R-loops formed

when an RNA:DNA hybrid displaces an unpaired DNA stand from a double-stranded DNA helix (Wu et al, 2022). The results indicate that the Z-DNA predictions for both Z-HUNT3 and Z-DNABERT align with only a small fraction of the ssDNA detected by K-seq. In Z-DNA forming regions, both strands undergoing modification.

The dotted boxes in Fig 2 highlight the different patterns of overlap between the approaches we examined. Panel a shows concordance of Z-DNABERT mappings by all approaches, panel b shows overlap of Z-DNABERT with K-seq, whereas panel c shows overlap of Z-DNABERT with both KEx and Z-HUNT3. The findings reaffirm that Z-DNABERT is not just capturing the KEx training set and that it can find hits predicted either by K-seq or by Z-HUNT3 (see also Fig S3A). Furthermore, the predictions also show differences with K-seq, which detects many ssDNA regions that exist independently of Z-DNA formation, such as those in the regions that lie between panels a, b, c, where strand-specific signals potentially arise because of R-loop formation. Furthermore, K-seq does not detect other signals where KEx and Z-HUNT3 maps overlap (Fig S3B). K-seq signals are reduced by triptolide and increased by DRB (5,6-Dichloro-1-b-D-ribofuranosylbenzimidazole) (Fig 2A). DRB also increases the overlap of signal with CG islands. The increase in intensity likely reflects the decreased presence of RNA polymerase in the gene body and does not necessarily indicate that more Z-DNA is present. The colocalization of NELF and AGO proteins highlights that Z-DNA signals are enhanced in proximal promoter regions as previously noted for the overlap of conserved microRNA seed sequences with flipons (Herbert et al, 2023).

Our Z-DNABERT results enabled us to ascertain whether certain motifs are enriched in BZ junctions, as junctional sequences are not used by Z-HUNT3 to predict Z-DNA formation and the KEx training set is seven times smaller in genomic coverage than the predicted set. We found that a d(TAAA) motif was enriched in the 5' region at both ends of the Z-DNABERT junction between B-DNA and Z-DNA (Fig 2C). The result is consistent with in vitro studies showing that adenosines form BZ junctions (Ha et al, 2005; Kim et al, 2018) and differs from that found using just the KEx dataset (Fig S3E). Furthermore, our finding supports the suggestion that some sequences do not favour BZ junction formation. Instead, those sequences can oppose the flip to Z-DNA by an otherwise Z-prone sequence (Kim et al, 2018).

In this article, KEx-tuned Z-DNABERT is used as the primary form of analysis. We retain the use of the other methods to validate findings from the detailed analysis of specific loci.

## Z-DNABERT and genomic features

We found that Z-flipons were widely dispersed through the genome (Fig 3A and Table S3). Around 30% of the predicted Z-flipons fell within promoters and were less than 1 kb from a TSS, with around 40% less than 3 kb distant. 30% are located in the introns with 7% found in the first introns and another 30% in intergenic regions (Fig 3B and C). The distribution differs from that found genome-wide (Fig 3D). The enrichment of Z-flipons in promoter regions is consistent with previous analyses (Champ et al, 2004) and existing experimental results (Shin et al, 2016; Kouzine et al, 2017). The maximum overlap of experimental Z-DNA versus predicted (95.32%) is observed in 5' exons < 300 bp from the TSS (Table 6) in transcriptionally active genes (Figs 2 and S3).

Overall, the predicted Z-flipon set incorporates 92% of KEx experimentally validated Z-DNA (Figs 3A and S4), but is seven times larger in size (290,071 versus the 39,766 segments of overlap, Table S2). Interestingly, we did not detect a substantial overlap with regions of G-banding or with high recombination frequencies, negating a number of previous proposals made without experimental support (Fig S5) (Rich et al, 1984). We do observe a marked overlap with subtelomeric regions that was quite unanticipated and is of interest given the role subtelomeric RNAs play in inducing ZBP1-dependent cell death during replicative stress (Nassour et al, 2023).

## Z-flipons are enriched in CTCF-bound proximal enhancer and promoter regions

We tested whether Z-flipons align with the candidate cis regulatory elements (cCRE) defined by the ENCODE Consortium (Encode Project Consortium 2012). The correspondence with CTCF (CCCTC-binding factor)-enriched sites at cCRE promoters is quite evident (Fig 3) and more pronounced than when each feature is considered separately (Fig S6). We explored the cCRE results presented in Fig 3B further. Almost 10% of the predicted Z-DNA fell into cCRE regions (91,292 out of 926,535). Specifically, enrichment was observed in CTCF-bound proximal enhancer (threefold enrichment) and promoter (6.7-fold enrichment) regions (Fig 3A and Table S3), consistent with a regulatory role for Z-flipons.

There were 393 of these transcription-associated cCRE regions where Z-flipons overlapped with variants identified by GWAS. Among them, 86 (22%) are editing quantitative trait loci (edQTL)

**Table 6. Genomic features of predicted and experimental Z-flipons.**

| Z-Flipons | | | | |
|---|---|---|---|---|
| | Predicted only | Experimental and predicted | Experimental only | Overlap of experimental with predicted |
| Promoter (<=3 kb) | 91,735 | 26,751 | 2040 | 92.91% |
| 5'UTR | 478 | 90 | 9 | 90.91% |
| Exons | 5,475 | 1,650 | 81 | 95.32% |
| Introns | 73,807 | 5,501 | 527 | 91.26% |
| 3'UTR | 1,641 | 325 | 23 | 93.39% |
| Downstream (<=300) | 234 | 32 | 2 | 94.12% |
| Distal Intergenic | 76,299 | 5,810 | 651 | 89.92% |

variants, 66 (17%) are expression QTL (eQTL), and 29 (7%) are splicing QTL (sQTL). Some of the reported edQTLs are more than 400 kb from an affected RNA-editing site (Table S4). Such a distance between associated elements raises the possibility that Z-flipons can act by altering the loop topology of chromatin domains to bring widely separated elements close together, facilitating their interaction (Dixon et al, 2012; Nora et al, 2012).

## Overlap of quantitative trait loci with Z-flipons

We overlapped predicted Z-flipons with disease-associated variants from the GWAS catalog (Fig 4 and Tables 7 and S4). We observed 3.2-fold enrichment of GWAS single nucleotide polymorphisms (SNPs) in Z-flipons. Out of 108,517 unique GWAS SNPs, 655 (0.6%) fell into Z-DNA regions. We compared experimental Z-DNA predictions with respect to overlap with GWAS variants, and found that Z-DNABERT predicts 95% (109 out of 115) variants from KEx. Expanding the GWAS-associated region by 500 or 1,000 bases on either side further increased the overlap with Z-DNABERT hits to 12,440 and 20,171, respectively (Table S4).

We examined the overlap of Z-flipons with GWAS variants that are also QTLs for editing levels, expression level or splicing (Table S4). Out of 661 total variants from GWAS-overlapping Z-DNABERT, 215 (33%) are edQTL, 149 variants (23%) are eQTL, and 78 variants (12%) are sQTL (Table S4). We explored GO enrichment of variant falling in Z-flipons and found enrichment in positive regulation of transcription from RNA polymerase II promoter (GO:0045944 FDR = $2.78 \times 10^{-4}$) and chromatin (GO:0000785 FDR = $7.89 \times 10^{-3}$), consistent with our other findings.

There was also a significant overlap of Z-flipons in the OMIM collection of Mendelian variants (Fig 4D) that we will discuss later as we develop the evidence for the flipon-dependent outcomes summarized in Fig 4E.

## Z-flipons in action: real world applications of Z-DNABERT

A natural question is to ask how flipon variants affect trait values. To answer this query, we investigated possible mechanisms through an extensive analysis of orthogonal databases. We were able to disprove many of the hypotheses tested by showing that they were incompatible with existing data. The analysis was robust as we could run many independent control experiments based on a large number of data points, something that is not possible with a single wet lab-based experiment. The curation we performed

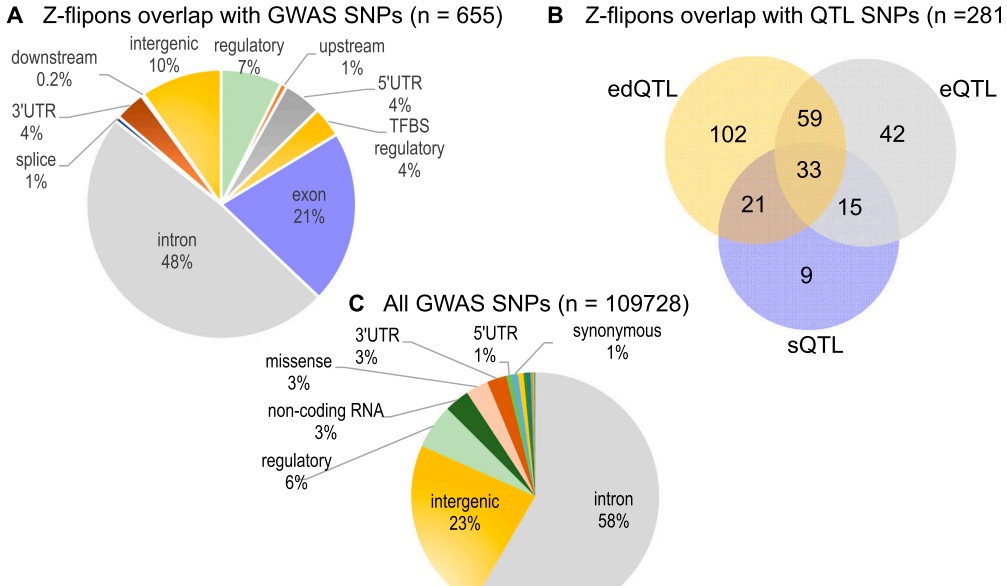

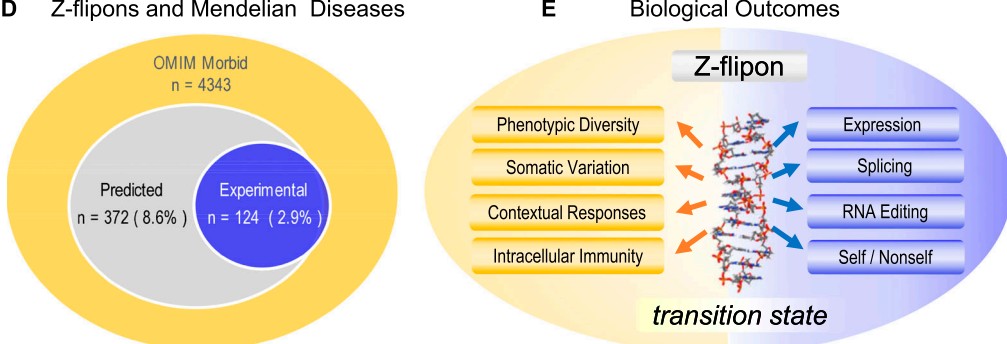

**Figure 4. Z-flipon overlaps with orthogonal genomic data.**
**(A)** Predicted Z-flipon overlaps with SNPs from the Genome-Wide Association Studies (GWAS) catalog. **(B)** Predicted Z-flipon overlaps with GWAS QTL SNPs. **(C)** Genomic features of all the GWAS SNPs analysed (features <0.5% are not labelled and are given in Table S7). **(D)** Genes in the (Online Mendelian Inheritance in Man) OMIM Morbid database with variants that overlap predicted and experimental Z-flipons. The predicted Z-flipon set captures 118 of the 124 genes with variants overlapping experimentally validated flipons. **(E)** The many ways that Z-flipons impact the phenotype by capturing and releasing energy to catalyse cell state transitions.

**Table 7. All GWAS SNPs by genomic features.**

| Context | Count | Percent |
|---|---|---|
| intron | 64,119 | 58.43% |
| intergenic | 25,459 | 23.20% |
| Regulatory region | 6,420 | 5.85% |
| Noncoding transcript exon | 3,585 | 3.27% |
| missense | 3,232 | 2.95% |
| 3'UTR | 2,775 | 2.53% |
| 5'UTR | 871 | 0.79% |
| Synonymous | 810 | 0.74% |
| TF-binding site | 808 | 0.74% |
| Upstream gene | 543 | 0.49% |
| Downstream gene | 412 | 0.38% |
| Splice region | 351 | 0.32% |
| Stop gained | 167 | 0.15% |
| Frameshift | 51 | 0.05% |
| Splice donor | 49 | 0.04% |
| Splice acceptor | 43 | 0.04% |
| Other | 33 | 0.03% |

allows targeting of future wet lab experiments to maximize both the efficient engagement of resources and the replicability of results.

Our results identify two novel Z-RNA repeat motifs that are likely involved in expression and splicing, both different from the conserved Alu Z-Box motif we previously identified as targeting A→I editing by the ADAR1 p150 isoform (Herbert, 2019b). The first motif has a Z-RNA stem associated with a loop containing an effector domain as we discuss in sections on *SCARF2*, *SMAD1*, and *CACNA1C* genes. The other Z-RNA motif overlaps a previously characterized intronic splicing enhancer sequence and has the potential to generate novel protein isoforms from zinc-finger gene arrays like those present on chromosome 19 (see *ZNF587B* section).

### An eQTL in *SCARF2* affects *MED15* and height

The rs874100 SNP (NM_153334.7:c.2459G>C), which encodes a non-synonymous variant (NP_699165.3:p.Gly820Ala) (Fig 5A), overlaps a predicted and experimentally confirmed Z-flipon (Fig 5B). The Z-DNABERT mutagenesis map reveals that the minor C allele disrupts Z-DNA formation (Fig 5C). The C allele also prevents the fold of the *SCARF2* transcript into Z-RNA (Fig 5D). The fold forms a loop anchored by the Z-RNA stem. A GU splice donor site at positions 89–90 is present in the loop, although there is no current evidence that the site is associated with alternative splicing.

The rs874100 SNP is an eQTL for the mediator complex subunit 15 (MED15) gene that is associated by GWAS with height. The microC map from human embryonic stem cells (hESC) reveals the presence of contacts between the rs874100 region and the *MED15* promoter (Blue Box, Fig 5E). We were able to define four haplotypes that

incorporate other neighborhood SNPs that are also associated with height (Fig 5F and G). The haplotypes also included the exon 7 nonsynonymous SNP rs2241230 (NM_153334.7:c.1273A>T variant (XP_016884554.1:p.Thr425Ser), which is not an eQTL but rather a sQTL and the intron 6 variant rs882745 (NM_153334.7:c.1203-97G>T) that is just upstream of an alternative splice site for *SCARF2* (Fig 5H). Although many of the SNPs do not overlap flipons, they help define haplotypes associated with high and low expressions of *MED15*.

The ZNA prone haplotype H1 is associated with increased expression of *MED15*, whereas haplotype H4 with the rs874100 C allele that disrupts the Z-DNA stem has low expression (Note that the lower strand is coding while the SNP allele is given for the top strand). This finding is supported by the two intronic SNPs rs1558170 (NC_000022.11:g.20433955C>G) and rs9610925 (NC_000022.10: g.20789046T>A) that are in strong linkage disequilibrium with rs874100. The increased MED15 gene expression of H1 relative to H4 could partly reflect the nonsynonymous changes produced by the SCARF2 SNPs rather than through differences in Z-DNA formation. This explanation is less likely as the rs874100 amino acid substitution has been shown by clinical testing to be benign (ClinVar accession RCV000602615.1). Furthermore, the variant produced lies in the disordered carboxy terminus of the protein and not within a functional domain. The other nonsynonymous SNP rs2241230 is not an eQTL for MED15 but a sQTL whose minor allele is associated with decreased splicing of MED15, likely offsetting the increased expression associated with the rs874100 G allele. The association of rs874100 with height may then reflect the higher expression of MED15 protein because of the formation of ZNAs by H1. The increased coupling between enhancers and promoters promoted by the MED15 mediator complex would increase cell growth by generating higher levels of transcripts and proteins. The altered splicing associated with rs2241230 may further affect MED15 expression levels by altering the isoforms produced.

### An eQTL in *SMAD1* affects *HDL* cholesterol

We observed a similar Z-RNA stem/loop motif in our analysis of eQTLs for *SMAD1*, a gene associated with cholesterol efflux from a cell (Feng et al, 2014). The eQTLs present in the 5'UTR of the *SMAD1* gene include rs13144151(A>G) (NC_000004.11:g.146403165A>G) and rs13118865(C>T) (NG_042284.1:g.5698C>T). The SNPs defined three haplotypes that express intermediate (H1), high (H2), and low (H3) levels of *SMAD1* mRNA. Both H2 and H3 contain potential Z-RNA-forming sequences. The high-expressing H2 incorporates the minor G allele of rs13144151 that overlaps an experimentally validated Z-DNABERT prediction (Fig 6D). Mutagenesis mapping of rs13144151 with Z-DNABERT revealed that G allele caused a slight increase in Z-propensity. Although not pronounced at the level of DNA, the effects of the allele on the RNA fold are quite evident (Fig 6G and H): the G allele stabilizes an additional potential Z-RNA helix by adding an extra G:C bp to increase its span to 6 bps, producing the minimal length substrate required to dock a Zα domain (Placido et al, 2007) (Fig 6G and H). The low-expressing H3 haplotype is defined by the minor alleles of rs13118865 and rs1264670 (G>A) (NC_000004.11: g.146402927G>A). Rs1264670 is incorporated into an RNA fold motif

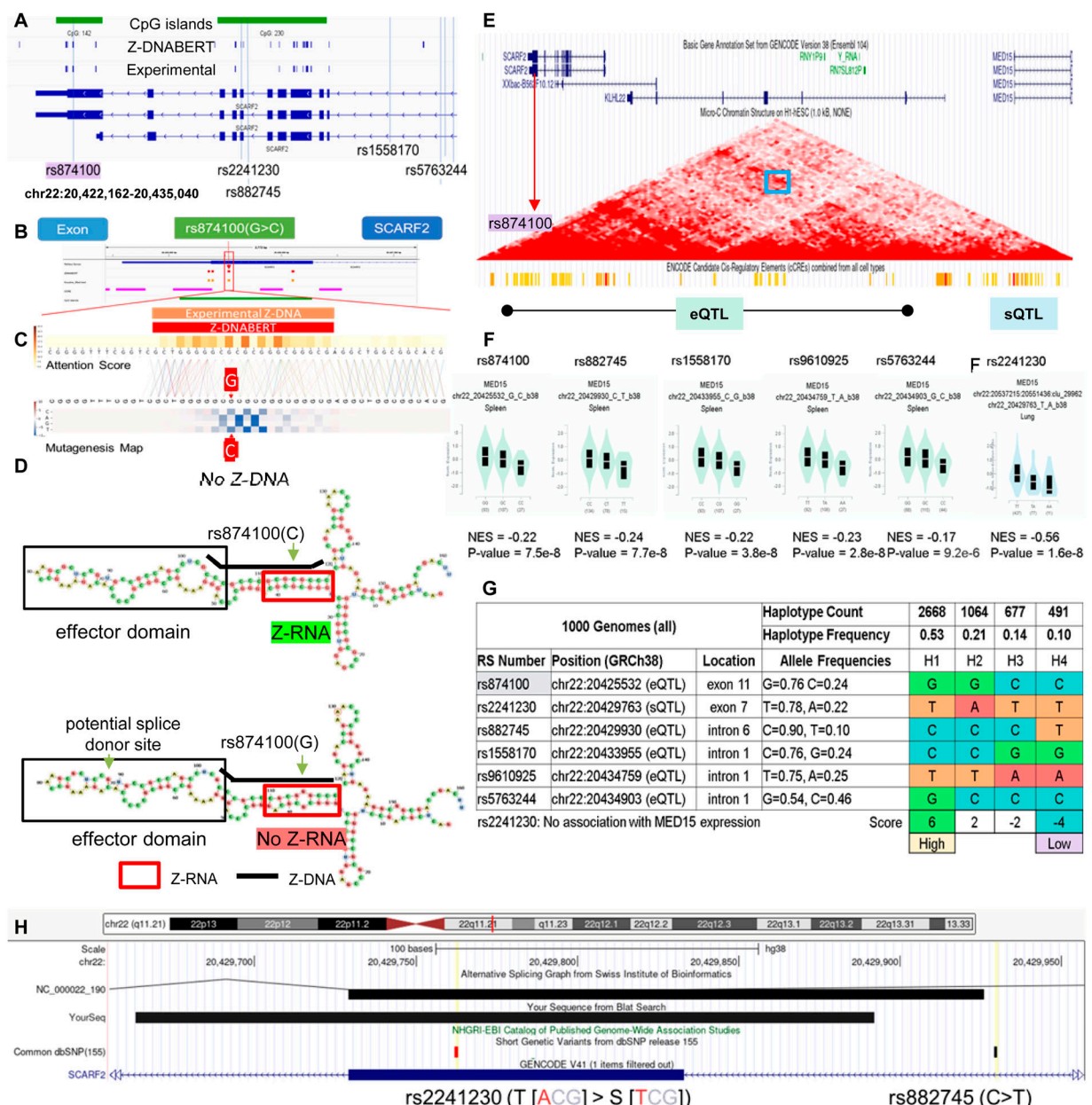

**Figure 5. A Z-flipon in *SCARF2* associates with *MED15* expression.**
**(A)** chr22:20,422,162–20,435,040 showing the 3' region of *SCARF2* along with SNPs used in the analysis. The positions of predicted and experimentally confirmed Z-flipons are also shown, along with CpG islands. **(B, C)** Overlap of eQTL rs874100 with Z-DNABERT prediction (C) Computation prediction of the effect of mutagenesis of each nucleotide in the Z-flipon region. The SNP variant A allele leads to loss of Z-DNA formation as indicated by the blue coloring. **(D)** The Z-RNA fold from the Z-DNABERT-predicted Z-DNA sequence is shown below the thick black line. A potential splice donor site is indicated although there is no evidence for its use. Note that the RNA is transcribed in the reverse direction from the genome. The SNP minor allele also disrupts the Z-RNA fold. **(E)** chr22:20,415,440–20,518,466 showing both *SCARF2* and *MED15* genes, along with a microC map from human embryonic stem cells, with the blue box highlighting the convergence of the red diagonals that indicate contacts between rs874100 and the *MED15* promoter. The bars show the cCRE in the *MED15* promoter that were mapped by the ENCODE consortium with orange indicating an enhancer, whereas red highlights a promoter region. **(F)** SNP eQTL for *MED15* showing the normalized effect size and *P*-value determined by the GTEX consortium. **(G)** The haplotypes were scored by adding +1 if a SNP allele was associated with an increase in trait value and −1 if the value was lower. Haplotype 1 favors Z-DNA formation and is associated with high *MED15* expression, whereas haplotype 4 has low expression of *MED15* and a low propensity to form ZNAs. **(H)** The rs2241230 SNP is positioned near an alternative splice site for *SCARF2* and is an sQTL for *MED15*.

similar to that of H1 with a Z-RNA stem and a hairpin loop domain (Fig 6I). Present in the domain are two unpaired splice donor sites and many CGGG sites of the type bound by the alternative splicing factor RBM4 (RNA-binding motif protein 4). Because RBM4 is known

to suppress the use of splice donor sites (Wang et al, 2014), we refer to the hairpin as an effector domain.

Interestingly, the SNP minor alleles affecting *SMAD1* expression map not only to haplotypes, but also to the exons defining different

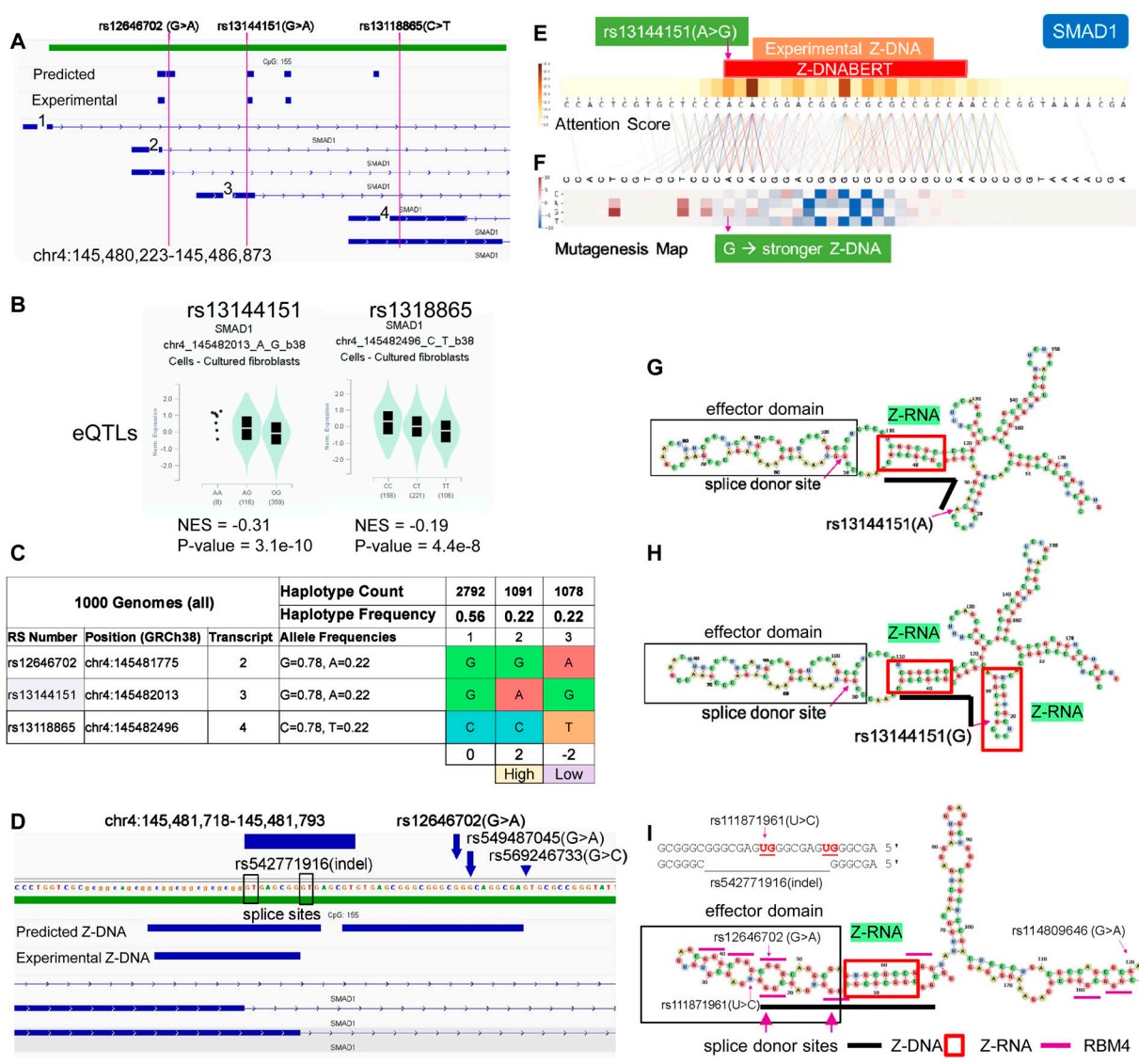

**Figure 6. *SMAD1* expression and splicing for rs13144151(A>G) and rs12646702(G>A).**
**(A)** Location of SNPs and splicing isoforms. The exons that are labeled 2, 3, and 4 are associated with different transcripts. After splicing, each transcript is uniquely marked by the presence or absence of a particular SNP in one of the numbered exons. **(B)** The rs13144151(A>G) and the rs13118865(C>T) SNPs affect the expression of *SMAD1* mRNA. No QTL data are available for rs12646702, but they are in linkage disequilibrium with rs13118865 that serves as a surrogate. **(C)** Haplotypes differ in their expressions of *SMAD1*. The haplotypes were scored by assigning +1 to the alleles that increased trait values and –1 otherwise. For rs12646702 where no quantitative trait information is available, both alleles were assigned a value of zero. **(D)** The 5′UTR of *SMAD1* in the vicinity of rs12646702(G>A) showing the Z-DNABERT-predicted Z-flipons and the experimentally mapped Z-flipons, SNP locations, and an alternatively spliced *SMAD1* exon. **(E)** Mapping at nucleotide resolution of the overlap of Z-DNABERT predictions and rs13144151. **(F)** Z-DNABERT predicted effects of nucleotide substitutions at this locus showing that the A>G substitution slightly enhances Z-DNA formation. Red indicates an increase in Z-propensity while blue represents decreased Z-DNA formation. **(G)** The Z-RNA stem and the effector domain loop containing a splice donor site formed in the vicinity of rs1314415. The heavy black line corresponds to the Z-flipon sequence predicted by Z-DNABERT. **(H)** The rs1314415 G allele enables formation of an additional Z-RNA stem that is associated with lower expression of the transcript. **(I)** Z-RNA-forming stem that includes rs12646702 is associated with an effector domain that contains CGGG-binding sites for the alternative splicing factor RBM4 indicated by short purple lines, with the heavy black showing the Z-flipon sequence. The SNP locations are shown along with the rs542771916 indel.

splice isoforms. The rs13144151 A allele that defines the H2 haplotype is present on exon 3, whereas H3 is defined by both the rs13118865 T allele on exon 4 and the rs1264670 A allele at the 3′ end of exon 2 (as labeled in Fig 6A). The strength of Z-RNA formation associated with each exon likely affects the expression of each isoform. The transcription of isoforms containing exon 3 may be favored by the rs13144151 A allele that disrupts Z-RNA formation and

allows RNA polymerase progression. In contrast, both exons 2 and 4 contain strong Z-RNA folds that could cause RNA polymerases, leading to lower readout of these isoforms.

The association of rs13144151 with HDL cholesterol levels (A allele = –0.018-unit decrease [Richardson et al, 2020b]) is consistent with the known role of *SMAD1* in negatively regulating cholesterol efflux from cells. Increased *SMAD1*

expression leads to decreased levels of the cholesterol transporters ABCA1 and ABCG1, with lipid accumulation by macrophages producing foam cells that are associated with atherosclerosis (Feng et al, 2014).

### A sQTL in *CACNA1C* affects *DCP1B* and BMI

We also assessed the relative roles of Z-DNA and Z-RNA in splicing by analyzing sQTLs found in the 5′ UTR of *CACNA1C* (calcium voltage-gated channel subunit alpha1 C) that alter the processing of decapping1B protein (DCP1B) transcripts. DCP1B protein initiates mRNA decay by enzymatically removing the 5′cap from RNAs. The index SNP rs11062091 (NG_008801.2:g.87418G>A) is an sQTL for *DCP1B* splicing but is not currently associated with a phenotype. Of the SNPs in the region nearby, rs2470397 (NG_008801.2: g.31165T>C) is a sQTL associated with BMI. In a GWAS of BMI in nearly half a million individuals (Zhu et al, 2020), rs10774018 (NG_008801.2:g.82974G>C) is an sQTL associated with visceral obesity and height (Karlsson et al, 2019; Richardson et al, 2020a) and rs2108635 (NG_008801.2:g.84605A>G) is associated with BMI but is not a QTL (Fig 7A). Haplotype analysis revealed that the major allele of rs11062091 is on haplotype H3, which scored highest for splicing, whereas the minor allele is on H6, which has the lowest score. In these haplotypes, the rs2470397 alleles do not correlate with those of other SNPs, most likely reflecting the high rate of recombination at this locus (Fig 7B). Nevertheless, the rs2470397 minor C allele helps define haplotypes 3 and 6 that evidence an association between the rs11062091 minor A allele, low levels of *DCP1B* splicing, and an increased incidence of obesity (Fig 7B). The effect on BMI may reflect the rate at which transcripts undergo decay, with H6 increasing the longevity of transcripts that promote fat accumulation.

The alternative *DCP1B* splice site and rs11062091 locus approximate each other as shown by the microC map generated from hESC. Both regions bear enhancer cCRE marks and overlap with CTCF-binding sites (Fig 7C and D). The sequences surrounding rs11062091 have many predicted and experimental Z-flipons, yet, Z-DNABERT mutagenesis maps revealed little effect of the SNP alleles on Z-DNA formation (Fig 7E). Analysis of the RNA fold revealed many regions of likely Z-RNA formation (red boxes) that did not align with experimentally validated Z-DNA (identified by heavy black lines). One of these contains a Z-RNA stem loop motif similar to those observed with *SCARF2* and *SMAD1* RNAs (Fig 7H).

With other Z-RNA stems, experimentally validated Z-DNAs aligned only with the upstream strand of the RNA and not its downstream complement. The only region where Z-DNA overlapped with both Z-RNA strands was the one that included rs11062091. The effect of the rs11062091 minor A allele was to disrupt the formation of this particular Z-RNA helical stem (Fig 7H). The results suggest that the Z-RNA incorporating rs11062091 nucleates the remaining RNA fold to facilitate splicing of the DCP1B transcript. The structures formed then serve to seed a spliceosome condensate. Indeed, rs11062091 is also a QTL for *RP5-1096D14.6* and *CACNA1C-IT2* splicing. The 12 canonical CCTC motifs in Z-RNA-associated effector domains could actively promote spliceosome formation by localizing CTCF to the region (Fig 7H) (Alharbi et al, 2021). Similar interactions may contribute to the

alignment of Z-DNA prone loci with CTCF/cCRE regions shown in Fig 3A.

The failure of Z-DNABERT to detect many of the Z-RNAs in this fold may reflect a limitation of the method that we discuss later.

### Z-flipons, edited and noncoding transcripts

Given the different requirements for Z-RNA formation compared with those for Z-DNA, we were interested to test how well Z-DNABERT identifies Z-flipons within genes that contain known sites of A→I RNA editing. We found that Z-DNABERT does not detect the Z-RNA-forming Alu sequencing known to impact ADAR1 editing of the cathepsin S (*CTSS*) RNA (Stellos et al, 2016; Nichols et al, 2021).

Overall, we found very few cases of direct overlap of Z-flipons with editing sites in the analysis of a number of published datasets (Table S5). As many editing substrates are long, we also searched for Z-flipons in 1 kb surrounding the editing site and found a higher overlap. One experimental study explored editing substrates recognized by the Zα domain of ADAR p150 (Sun et al, 2021). Of the 1,248 mRNAs identified, none had a Z-DNABERT overlap. Expanding the search window for a Z-flipon prediction to 1 kb revealed that an overlap with only 4% of the ADAR1 p150-editing sites. A separate study of lung adenocarcinoma tumors (Sharpnack et al, 2018) found 1,413 genes where the total level of RNA editing and expression were correlated. Of these, 5% of edited sites have a direct overlap with Z-flipons (Table S5). Expanding the region of search for Z-flipons within 1 kb of editing sites increased the overlap to 19%. We further found that 182 of the transcripts immunoprecipitated with the ZNA-specific antibody Z22 from mouse embryonic fibroblasts (Zhang et al, 2022) (Table S5), providing an experimental confirmation for Z-DNABERT prediction with results suggestive of Z-flipon conservation between mouse and human. For the ADeditome database, which maps 1,676,363 editing sites associated with Alzheimer's disease (Fig S7), only 271 overlapped a Z-flipon prediction, of which 6 were validated experimentally (Table 8). However, Z-flipons were found within 1 kb of editing sites in 50% of ADeditome genes, a colocalization much higher than expected by chance (Table 9). It is uncertain how many of these edits are antemortem.

The cases where we were able to overlap Z-flipons with editing sites were for Z-RNA stems 12 bp or longer (Figs S8 and S9). For example, the intronic dsRNA of the potential negative regulator of insulin secretion, syntaxin-binding protein 5L (*STXBP5L*) (Bhatnagar et al, 2011) is short and heavily edited (Fig S8). In contrast, only a single edit (reproduced in the lung adenocarcinoma dataset (Fig S9 and Table S5 and in the Rediportal database) is present in the *BIRCA* transcript, raising the question of whether the edit is functional. Interestingly, this editing site is within the site matching *hsa-miR-8485*, *hsa-miR-574-5p*, and *hsa-miR-297* microRNA (miR) seed sequences. These miRs target TDP-43 (encoded by *TARDBP*) to repress *NRXN1* expression (Fan et al, 2014). Our finding raises the possibility that ADAR1 p150 dependent editing of *BICRA* RNA regulates its suppression by TDP-43. Alternatively, the edit could indicate that the RNA is engaged by ADAR1 p150 but the outcome of the interaction is not editing dependent. Another instance where Z-RNA may enable regulation of noncoding RNAs is provided by *RMRP* (RNA component of mitochondrial RNA processing endoribonuclease)

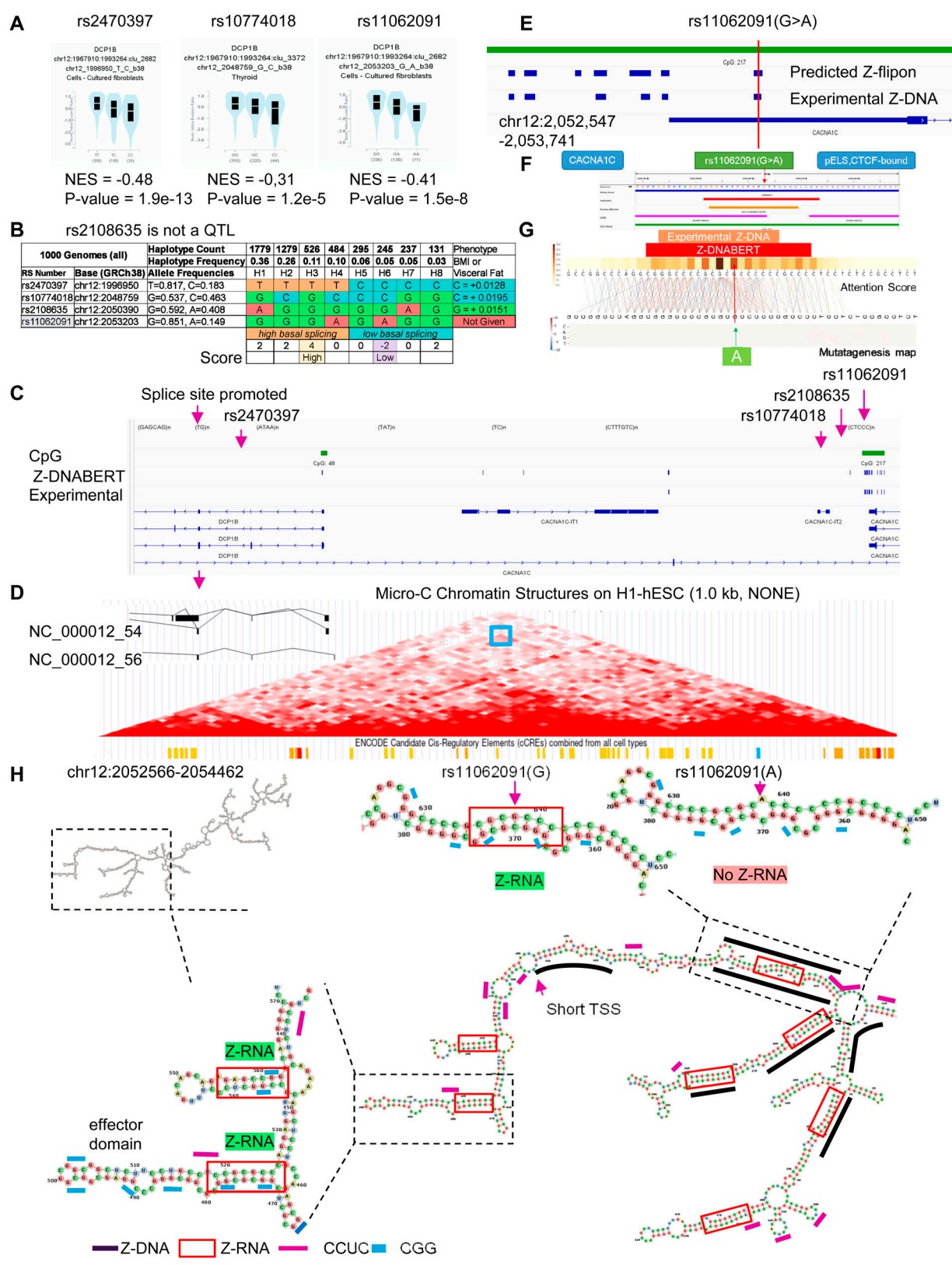

**Figure 7. A Z-flipon in *CACNA1* associates with upstream *DCPB1* expression.**
**(A, B)** Z-flipons in (hg38.chr12:1,935,235–2,714,656) in *CACNA1* (calcium voltage-gated channel subunit alpha1 C) affect splicing of *DCP1B* (decapping mRNA 1B). **(A)** Minor SNP alleles are associated with decreased splicing of *DCP1B* transcripts. **(B)** Haplotype map of the region that supports an association between decreased *DCP1B* splicing and increased body mass index. The haplotypes were scored by adding +1 to the total if the allele was associated with an increase in trait value and −1 if the value was

**Table 8. Direct overlap of ADeditome edits with predicted Z-flipons.**

| Location | ADeditome | ADeditome and Z-DNABERT | ADeditome percent |
|---|---|---|---|
| Downstream | 1,805 | 1 | 0.11% |
| Exonic | 3,686 | 5 | 0.22% |
| Exonic; splicing | 9 | | 0.00% |
| Intergenic | 17,501 | 4 | 1.04% |
| Intronic | 1,413,894 | 226 | 84.34% |
| Noncoding RNA exonic | 23,022 | 3 | 1.37% |
| Noncoding RNA exonic; splicing | 37 | | 0.00% |
| Noncoding RNA intronic | 123,445 | 18 | 7.36% |
| Noncoding RNA splicing | 72 | | 0.00% |
| Splicing | 255 | | 0.02% |
| Upstream | 618 | | 0.04% |
| Upstream; downstream | 111 | | 0.01% |
| 3′ UTR | 87,697 | 9 | 5.23% |
| 5′ UTR | 4,116 | 5 | 0.25% |
| 5' UTR; 3' UTR | 95 | | 0.01% |
| Total | 1,676,363 | 271 | 100% |

**Table 9. ADeditome genes with predicted Z-flipons within 1 kb of an A →I edit.**

| | ADeditome edited genes | Z-Flipon (±1 kb of editing site) | % |
|---|---|---|---|
| Gene Count | 14,288 | 6,552 | 45.86% |

(Fig S10). The RNA can act as a sponge for miR, although it also has other roles in the nucleus and mitochondria (Hussen et al, 2021). The Z-prone sequences predicted by Z-DNABERT potentailly fold into Z-RNA stems able to engage ZBP1 and activate cell death when released by damaged mitochondria.

### ZNF587B and RNA editing

A predicted and experimentally validated Z-flipon within the *ZNF587B* gene that is associated with many nonsynonymous edits led us to investigate the locus further (Fig 8). The gene is in one of the zinc finger (ZNF) gene clusters enriched on chromosome 19 (Figs 8 and S11). Depending on how it is spliced, *ZNF587B* contains up to 13 zinc finger domains (ZNF), plus a KRAB (Krüppel-associated box)

domain of the kind thought to mediate the repression of transposon repeat elements (Ecco et al, 2017). *ZNF587B* RNA editing is promoted by a number of Alu inverted repeats (AIR) similar to those of known ADAR1 substrates (Fig 8A). They overlap the terminal exon of one RNA isoform and result in RNA recoding specific to that transcript (Fig 8B and C). A different type of RNA fold directs editing of the other *ZNF587B* splice isoform (Fig 8B). Interestingly, the dsRNA in this region forms from heptamer repeats (HR) that create clusters of unpaired RNA loops distinct from the long, linear AIR substrates (Fig 8C–E). The HR has purine–pyrimidine inverted repeats capable of forming short Z-prone dsRNA helices (Placido et al, 2007; Nichols et al, 2021) similar to those clusters we recently identified in mouse by immunoprecipitation with ZNA-specific Z22 antibody (Zhang et al, 2022).

lower. The highest and lowest scores are associated with rs11062091 alleles. **(C)** Location of the alternative *DCP1B* splice along with the position of all SNPs. **(D)** The alternatively spliced *DCP1B* transcript is drawn as an inset to the microC map that shows that contact is present between the SNP locus and the *DCP1B* genic region, as indicated by the region boxed in blue. The areas of contact contain chromatin modifications classified as cCRE by the ENCODE project (orange bars represent enhancers and red bars are for promoters). SNP positions, simple repeats, and both predicted and experimental Z-DNA are shown. The *CACNA1C* splice site affected by rs11062091 is upstream (chr12:1,967,910–1,993,264) and is currently not annotated in GENCODE 41. **(E)** Expanded view of Z-DNA in the vicinity of rs11062091 showing the overlap between the Z-DNABERT-predicted and experimentally validated Z-flipons. **(F)** Z-DNABERT prediction for the Z-flipon that incorporates rs11062091. **(G)** Z-DNABERT mutagenesis shows that single nucleotide variants do not affect the propensity of the rs11062091 Z-flipon to form Z-DNA as the heat map does not change with base substituion. **(H)** Progressively zoomed in views of the dsRNA fold of the transcript from the rs11062091 region. The A allele of rs11062091 disrupts formation of Z-RNA. The black lines show the experimentally determined regions of Z-DNA formation. Only the rs11062091 Z-flipon experimentally forms Z-DNA at the locations where the two RNA strands that create the Z-RNA stem are transcribed from. Multiple Z-RNA prone helices are formed with RNAs transcribed from regions where Z-DNA formation was not experimentally detected. The short purple lines show CCUC motifs that could represent CTCF protein-binding sites. The RNA fold overlaps the transcription start site (TSS, chr12:2,052,986) for the shorter *CACNA1C* transcript as indicated by the TSS label. A Z-RNA stem/loop effector domain motif resembling those in Figs 5 and 6 is also illustrated with short blue dashes above CGG repeat sequences.

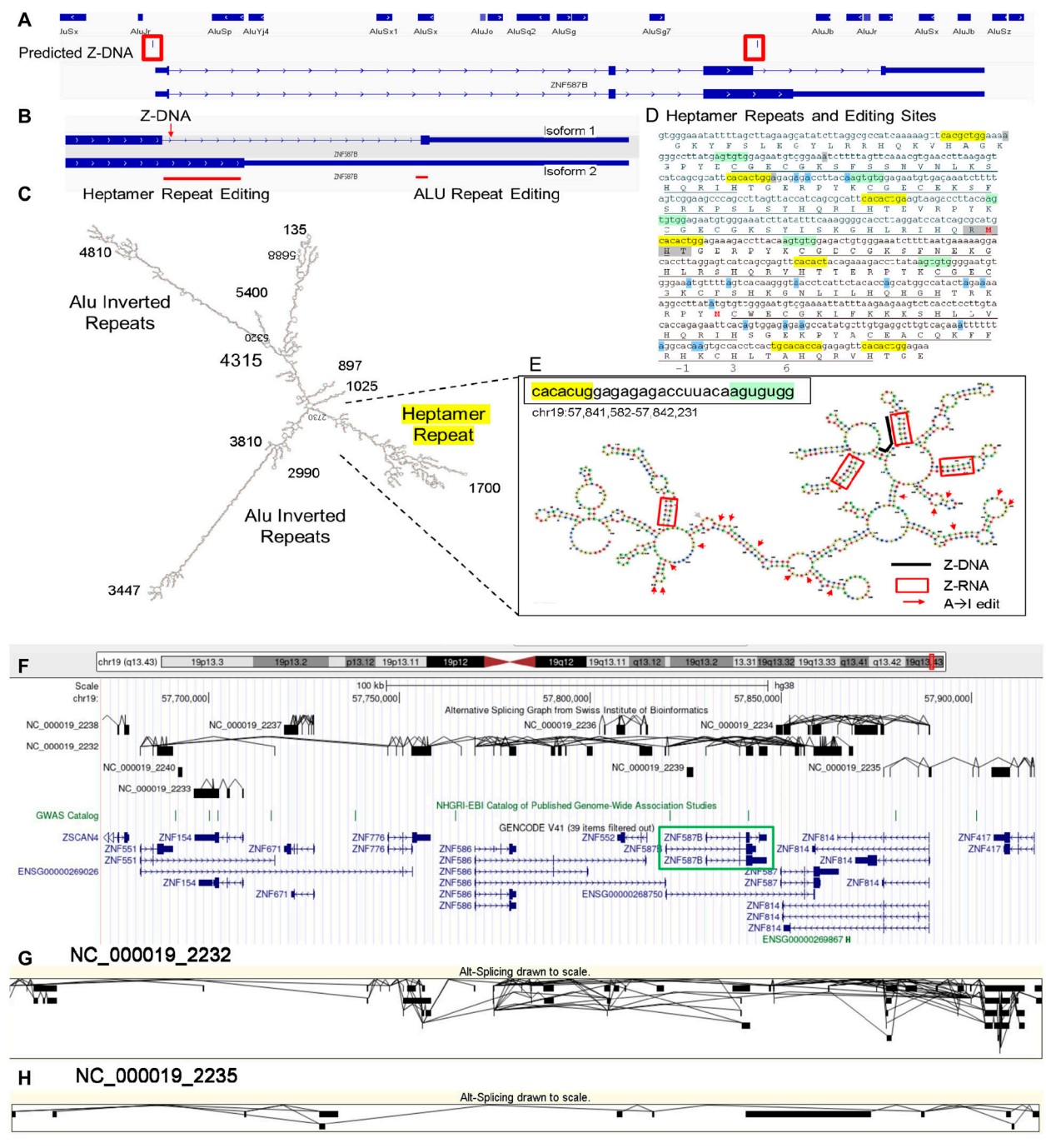

**Figure 8. Nonsynonymous RNA editing of *ZNF587B*.**
**(A)** *ZNF587B* locus. **(B)** *ZNF587B* isoform-specific RNA edits occur in different exons. **(C, D)** dsRNA fold showing two classes of editing substrate (D) dsRNA region maps to C2H2 zinc finger (ZNF) repeats that have a $CX_{2-4}CX_{12}HX_{2-6}H$ motif (X is any amino acid) and are underlined. The ZNF domains are joined by a seven amino acid linker that is within the heptad repeat. The gray box lies underneath the Z-DNABERT-predicted Z-DNA sequence and the blue boxes highlight residues with nonsynonymous edits. **(D)** The numbering immediately below the sequence in panel (D) corresponds to the DNA-binding residues of the α-helix of the ZNF above. **(E)** Heptad repeat folds are highlighted and the Z-RNA-prone sequences are within the red boxes. The arrows indicate A→I editing sites. The heavy black line is above the predicted and experimentally validated Z-flipon sequence. **(F, G, H)** Alternative splicing within chromosome 19 telomeric zinc finger gene cluster (hg38 chr19:57,672,145–57,921,020) with two of the trans-splicing isoforms displayed in (G, H).

The length of the HR is conserved. It encodes the linker between adjacent ZNF (Fig 8D). Interestingly, the CACA motif overlaps that of known intronic splicing enhancers (Deletang & Taulan-Cadars, 2022), raising the possibility that Z-RNA formation by the HR modulates alternative splicing. The arrangement of ZNF in clusters may enable intergenic splicing to generate new combinations of ZNFs at the RNA

level. Evidence for the alternative splicing and trans-splicing from the Swiss Institute of Bioinformatics-curated dataset is shown in Fig 8F–H.

The generation of these novel transcripts would be favored by the interferon induction of the known Z-RNA-binding proteins ADAR1 p150 and ZBP1. The nonsynonymous edits scattered through the fold are consistent with Z-RNA-dependent localization of p150 to these transcripts. None of the edits alter the three residues (called −1, 3, and 6 as numbered on the bottom line of Fig 8D) that are involved in DNA recognition by ZNF (Munro et al, 2018), so they do not change the specificity of the ZNF. The altered splicing rather than RNA editing may be the major outcome produced by ADAR1 p150 as binding of p150 to the Z-RNA helix would occlude the site and make it unavailable to the splicing machinery. Alternatively, the interaction could help direct the locus to a spliceosome condensate of ribonucleoproteins that process the pre-mRNA. The novel combinations of ZNF produced by alternative splicing could prevent the escape of recently recombined transposons and viruses from KRAB-mediated suppression.

### Z-flipons, Mendelian disease, and LOF variants

We also examined Z-flipons for association with Mendelian disease (Table S6 and Figs S12–S20) given the previous emphasis placed on Z-DNA as a cause of genomic instability (Wang & Vasquez, 2014). There is overlap between experimentally determined Z-flipons and Mendelian variants in a number of genes including *HBA1* (hemoglobinopathies), *CDKN2A* (melanoma susceptibility), *MC1R* (red hair color, melanoma), *WNT1* (osteogenesis imperfecta, type xv), *NPHS1* (nephrotic syndrome, Type 1), *SOX10* (Waardenburg syndrome, Type 2e), *IDUA* (Hurler–Scheie syndrome), *LAMB3* (heterotaxy), *IL17RC* (familial candidiasis), and *FOXL2* (blepharophimosis, ptosis, and epicanthus inversus, type I), providing direct evidence that Z-flipons do influence trait variation. Predicted Z-flipons also overlap with a more extensive range of OMIM phenotypes. Examples include *TERC*, the telomerase RNA, *TERT*, *TP53*, *LMNA*, *NKX2.5*, *HBA2*, and *NROB1*. Overall, we found an overlap of Mendelian disease-causing variants with predicted (n = 372) and experimentally validated (n = 124) Z-flipons in 8.6% and 2.9% of OMIM genes (n = 4,343), respectively (Fig 4D). Most of the events (71%) with experimentally validated Z-flipons were because of nonsynonymous variants that altered arginine codons in 22% of cases (Figs S21 and S22), whereas 22% of variants were LOF frameshifts (Fig S23). We also analyzed the 430,056 predicted LOF (pLOF) variants listed in the Genome Aggregation Database (gnomAD) that are distributed over 19,012 unique genes (Karczewski et al, 2020). Of these, 4,362 variants fell into predicted Z-flipons. Interestingly, of the 1,160 variants present in the KEx dataset, 1,093 (94.2%) are in the gnomAD-pLOF set. Frameshift deletions were also more frequent with Z-flipon overlaps compared with other Z-flipon LOF classes and compared with the entire gnomAD-pLOF variant collection (Fig S23 and Table S7). Overall, 637 of the 2,614 Z-flipon LOF genes (24.7%) overlapping the gnomAD-pLOF have OMIM morbid phenotypes (n = 4,343), compared with 21.1% of the gnomAD-pLOF genes present in OMIM. Interestingly, the overlap of the Z-flipons present in the gnomAD-pLOF with OMIM genes is much higher than the actual number of Z-flipons recorded in OMIM. There is a 14.7% overlap of genes with gnomAD-pLOF-predicted Z-flipon variants and a 3.9% overlap with genes containing experimentally validated Z-flipons (Fig S24 and Table S7). GO analysis of Z-flipon Mendelian variants annotated in OMIM showed enrichment for transcriptional activity, homeobox proteins, and transforming growth factor regulators of the extracellular matrix (Table S6).

## Discussion

Discovering the functional roles of Z-flipons and mapping the associated phenotypes is a challenging task, as previously noted (Morange, 2007). The energy necessary to power the flip from B-DNA to Z-DNA inside cells can be generated by RNA polymerases and helicases during transcription or by the ejection of nucleosomes from DNA. Flipon base modifications or interactions with noncoding RNAs can further modulate the dynamics of Z-DNA formation (Herbert, 2022; Herbert et al, 2023).

We used genome-wide data and computational experiments to genetically map flipons to QTLs and disease outcomes. We used a machine learning approach called Z-DNABERT to detect Z-flipons by tuning the transformer algorithm implemented in DNABERT (Ji et al, 2021) with experimentally validated Z-DNA-forming sequences obtained from the human genome. By conjointly using a variety of predictive and experimental methods, we were able to show Z-flipons are enriched in promoters where they can catalyze the turnover of protein complexes involved in transcription (Herbert, 2022).

The Z-DNA-forming regions were detected experimentally through reagents such as KMnO4 and kethoxal that detect unpaired bases. The enrichment we find in promoters occurs in regions where the high levels of negative supercoiling detected by other means (Kouzine et al, 2013; Teves & Henikoff, 2013; Georgakopoulos-Soares et al, 2022) are sufficient to induce a flip from B-DNA to Z-DNA. The KEx approach was designed to partition the ssDNA regions detected into those associated with RNA polymerase transcription and those in which NoB-forming sequences were associated with higher-than-expected $KMnO_4$ modification. This method is validated by the strand-specific K-seq results presented in Fig 2 that were designed to detect R-loops formed during active transcription. In contrast to R-loops, Z-DNA formation produces modifications either to both DNA strands in a region or is not associated with R-loop formation.

Our findings also suggest that certain motifs favor the formation of BZ junctions. Consistent with in vitro studies, we detected a strong preference for adenosines at a BZ junction (Ha et al, 2005; Kim et al, 2018). The result stems directly from the Z-DNABERT algorithm as Z-HUNT3 is agnostic to the BZ junction sequence and assigns the same penalty to all. Yet, we saw a genome-wide enrichment of adenosines at BZ junctions with the exclusion of other bases. The strongest motif found by Z-DNABERT (d(TAAA) at the 5′ end of the junction (Fig 2C) was not apparent in the KEx dataset (Fig S3E), even though ~15% of segments were common to both sets. This type of BZ junction is likely favored at the ends of the d(AC)$_n$ repeats that are also enriched in the same motif.

The repeat adenosines in a BZ junction will likely affect the local DNA conformation. In crystal structures, the BZ junction has an 11°

bend that could be extended by the additional adenosines we find present in the in vivo data (Hizver et al, 2001). The conformational flexibility of a BZ junction that results from the additional adenosines may also facilitate intercalation by small molecules and the docking of high-mobility group proteins that preferentailly bind this motif (Bewley et al, 1998; Strahs & Schlick, 2000). The biological effects of this class of B-Z junctions require additional investigation.

We were also able to show enrichment of Z-DNA-forming elements in LINEs and simple repeats relative to genomic levels, suggesting that these sequences are subject to positive selection, indicating that they likely enable useful adaptations. We also recovered SINEs that were excluded from the KEx dataset during its preparation (Kouzine et al, 2017). Overall, there was enrichment of Z-flipons in promoters. The presence of multiple flipons in many promoters suggest that promoter conformation is quite variable with the partitioning of energy between B- and Z-DNA leading to many different shape combinations (Herbert, 2022). The potential of repeats like d(CGGG)$_n$ to adopt both Z-DNA and G4-quadruplex conformation adds an additional level of complexity. G4 formation is likely favored when there is sufficient energy to form the additional B-NoB junctions required. The data we analyzed come from a population of cells and likely do not capture the full variation present in each cell or any context-specific effects arising from differences in expression of coding and noncoding transcripts. The work here provides a roadmap for the further experimental exploration at the single-cell level. Although flipon sequences have low intrinsic informational value because of their high frequency in the genome, they can affect a wide range of phenotypes by adopting an alternative DNA structure (Herbert, 2019a).

We also provide an estimate for the number of genes where Z-flipon variants are causal for Mendelian diseases by starting with experimentally validated Z-DNA-forming sequences and using these results to predict additional Z-flipons in the genome. We found a direct overlap between Mendelian disease-causing variants with predicted (n = 372) and experimentally validated (n = 124) Z-flipons in 8.6% and 2.9% of OMIM genes (n = 4,343), respectively (Fig 4D). This conservative approach misses those OMIM genes where the variants impacting Z-flipon biology are not in the region of overlap. The LOF alleles identified were enriched for frameshifts, with homeobox genes and other transcriptional regulators showing increased susceptibility (Table S6). The flipons involved are likely those prone to freeze in the left-handed conformation either because of their length or location in genomic regions of high topological stress, resulting in DNA breaks and error-prone repair that increases the frequency of variation. Such events may be prevalent early in development when cell cycles are as short as 3 h and hypertranscription is prevalent (Percharde et al, 2017). Despite the low frequency of their occurrence, the Z-flipon, LOF variants may produce Mendelian diseases more often than more common causes of DNA damage because they induce frameshifts with higher penetrance.

We identified additional LOF variants that overlap Z-flipons in the predicted gnomAD-pLOF collection, but many are not currently associated with Mendelian disease (Fig S24). Their negative impact may be lessened by alternative splicing, as variants affecting splice sites are more frequent in gnomAD-pLOF (Cummings et al, 2020)

than we observe with direct OMIM Z-flipon overlaps. Other mechanisms such as transcript destabilization, nonsense-mediated RNA decay, and limited or tissue-specific expression also could play a role. In addition, it is likely that many pLOF variants are somatic rather than germline (Wiktor-Brown et al, 2006; Herbert, 2008). Z-flipons also overlap nonsynonymous variants that produce Mendelian diseases. Around 22% affect arginine codons that contain the Z-prone CG dinucleotide. Yet, there is no evidence that these codons are replaced by the alternative less Z-prone AGA or AGG arginine codons, even though the *HBA1* locus clearly demonstrates the possibility of wide-ranging codon replacements in Z-flipon sequences (Table S6 and Figs S12 and S21), suggesting that Z-flipon-forming sequences are of sufficient biological utility to conserve.

Z-DNABERT was also helpful in finding Z-RNAs but is not optimized for detecting Z-RNA. This limitation may account for the low frequency of predicted Z-prone sequences in Alu SINEs (Table 4). We and others have demonstrated experimentally that a Z-Box enables Alu inverted repeats to form a Z-RNA helix shorter than 12 nucleotides long and that can incorporate noncanonical base pairs (Herbert, 2019b; Nichols et al, 2021; Zhang et al, 2022). However, no SINEs were available to tune Z-DNABERT performance as they were removed from the ssDNA-seq results during the creation of the KEx dataset (Kouzine et al, 2017). There may be other more basic reasons for why Z-RNA-forming sequences were not detected by Z-DNABERT. When compared with Z-DNA, a major difference in the requirements for Z-RNA formation is in the energy expended in establishing the junction between left- and right-handed helices. With Z-RNA, loops and mismatches facilitate formation of a junction by lowering the energy cost. Z-DNABERT does not take account of these differences between Z-RNA and Z-DNA. It is not trained to look for the features favoring Z-RNA formation as it defaults to a Z-helix of 12 bp and does not search for complementary sequences that might pair to form Z-RNA. Despite these limitations, we were able to use the ability of Z-DNABERT to find the 5′ stem of a potential Z-RNA-forming sequence and perform computational mutagenesis to distinguish between Z-DNA- and Z-RNA-dependent events.

We found that other effects of Z-flipons at promoters in normal cells likely occur at the level of Z-RNA and involve motifs that have a Z-RNA stem paired with a hairpin loop containing an effector domain. One such example in *SMAD1* RNA is characterized by RBM4-binding motifs that promote alternative splicing by suppressing use of splice donor sites. Similar motifs with different effector domains were present in *SMAD1*, *SCARF2*, and *CACNA1* RNAs. We found examples where disruption of a Z-RNA stem by a SNP allele was associated with the reported GWAS phenotype.

We identified a different motif based on a 7 nucleotide HR present in Krüppel-associated box (KRAB) domain containing ZNF-containing proteins (pZNF), many of which are clustered on chromosome 19 (Figs 8 and S11). These pZNF bind to relatively conserved sequences in transposons and viruses and suppress their expression. Together, this family of proteins constitutes an intracellular form of immunity that protects against these invasive elements (Ecco et al, 2017). Here, we provide evidence that this system is adaptive, with the HR used to generate pZNF with novel combination of ZNFs, allowing recognition of more diverse targets.

The HR in this family of proteins is found between adjacent ZNFs (Iuchi, 2001); for example, *ZNF587B* RNA codes for a protein with 13 C2H2 (two cysteines, two histidines, Fig 8) The HR sequence has some remarkable properties. It can base pair with another HR to form a Z-RNA stem (Fig 8E). In addition, the HR composition resembles that of a recombination recognition sequence (RSS) similar to the one cleaved by RAG1 during immunoglobulin gene rearrangement (De et al, 2004). Furthermore, the repeat sequence has a strong match to a previously characterized intronic splice enhancer (Deletang & Taulan-Cadars, 2022). These HR features foster multiple mechanisms for thwarting transposons and viruses that escape suppression by rearranging the binding sites targeted by existing pZNFs. At the DNA level, a protein like RAG could create new ZNF arrays through site-specific recombination based on HR as occurs in B- and T-cell receptor genes. We did not find evidence for an increased rate of indels or gene fusions associated with ZNFs in cancer datasets, especially in liver tissues where stellate cells express high levels of RAG1. However, there is an elevated level of missense mutation in some cancer types at ZNF positions 9 and 11 (Munro et al, 2018) adjacent to the HR "ACA" sequence, similar to that found in RAG1 cleavage substrates. Possibly such RAG1-dependent recombination of HPs occurs over longer time periods to diversify ZNF arrays and may account for the clusters that are now present on chromosome 19 (Fig S11). Consistent with this possibility is our observation that 179 of 252 degenerate ZNFs listed in UNIPROT are found in the KRAB domain containing C2H2 ZNF family.

Interestingly, the unique chromatin structure of C2H2 ZNF clusters reduces DNA recombination of these regions by localizing the H3.3 variant to ZNF-containing exons through interactions dependent upon *ZNF274* and the ATRX chromatin remodeling complex (Frietze et al, 2010; Valle-Garcia et al, 2016; Timpano & Picketts, 2020). At the same time, alternative splicing in this region is favored by the increased levels of H3K36me3 present (Luco et al, 2010). A similar chromatin structure is present at telomeres and also decreases recombination. Interestingly, the same structure is also found at the *HBA1* locus (Ratnakumar et al, 2012; Truch et al, 2022). Taken together, the findings raise the possibility that this unique chromatin structure enhances evolutionary adaptation by allowing rapid variation in rates of DNA recombination and RNA processing of the associated genes. The diversity of outcomes produced increases the probability that some individuals will survive when an existential threat emerges. Malaria is one pathogen that drives *HBA1* variation (Thom et al, 2013), although alternative telomere maintenance in cancer cells through enhanced recombination of chromosomal ends proves another example of how effective this mechanism can be in generating diversity (Bryan et al, 1997). The high frequency of flipons in subtelomeric regions suggests these regions are cynosures of evolutionary adaptations (Fig S5C).

In contrast to DNA-mediated events, generating pZNF variation at the RNA level is a much more rapid process (Herbert, 2004). Although RNA editing recodes ZNF, we did not find nonsynonymous edits that affected the key ZNF nucleic-binding sites. Instead, we found evidence supporting the possibility of an adaptive system based on trans-splicing within ZNF gene clusters, possibly by occlusion of HR splice enhancer sites by proteins engaging them as Z-RNA. Such RNA recombination events do not change the

specificity of the ZNF but generate new permutations to match a novel transposon or viral rearrangement. Those that enable a cell's survival likely will be fixed in that cell by epigenetic modifications. Alternatively, they may be fixed by reverse transcription (Herbert, 2004), possibly using a cleaved HR as a primer to embed the new ZNF combination in an existing ZNF gene.

The results we describe here are consistent with a model where ZNAs localize proteins to a site where they act. In the nucleus, Z-DNA can catalyze the turnover of the cellular machinery at promoters to regulate gene transcription as evidenced in a number of previous studies (reviewed in Herbert [2021a] and [2022]). The chromatin structures and condensates formed, some of which may be based on the Z-RNA motifs described here (Figs 5–7), enable approximation of distant regions through loop formation independently of loop extrusion (Fudenberg et al, 2016). They also help maintain nuclear structure in the absence of cohesin (Schwarzer et al, 2017). The formation at the same locus of both Z-DNA and Z-RNA may increase the efficiency of RNA processing. Z-DNA may initially localize the cellular machinery to sites of active transcription. The proteins then are positioned to bind Z-RNA formed by folding of nascent transcript. Such a process would enable nuclear editing of pre-RNAs by ADAR1 p150 and the interactions with the noncoding RNAs involved in splicing. Cytoplasmic formation of Z-RNA during infection, inflammation or as a result of cellular stress can enhance or inhibit immune responses (Herbert, 2020b).

Overall, we associate experimentally validated Z-flipons with active promoters that we then link to quantitative and disease phenotypes through the analysis of orthogonal genome-wide datasets. The work furthers our understanding of flipon biology and establishes a community resource. The hypotheses generated are data driven and open new lines of enquiry into the germline and somatic mechanisms that lead to QTL variation and disease.

# Materials and Methods

### Experimental Z-DNA training data

Permanganate/S1 Nuclease Footprinting Z-DNA data contained 41,324 regions with a total length of 773,788 bp in humans. We downloaded the dataset "ssDNA + SMnB" from https://www.ncbi.nlm.nih.gov/CBBresearch/Przytycka/index.cgi#nonbdna (Kouzine et al, 2017). We verified the mapping to hg19 and filtered out ENCODE-blacklisted regions. The ssDNA wiggle file was downloaded from the same location. For DNABERT, the data were preprocessed by converting a sequence into 6-mer representation. Each nucleotide position is represented by a k-mer consisting of a current nucleotide and the next five nucleotides. The data were split into five stratified folds so we could train five individual models with 80% of the data and assess precision and recall using the remaining 20%. Because of the large imbalance between positive (Z-DNA) and negative (not Z-DNA) classes, we randomly sampled twice as many of the negative class from the Kouzine et al human data.

### Deep learning transformer-based model training

DNABERT was fine-tuned for the Z-DNA segmentation task with the following hyperparameters: epochs = 3, max_learning_rate = 1 × $10^{-5}$, learning_rate_scheduler = one_cycle (warmup 30%) batch size = 24. We trained five models, each on 80% of the positive class examples, and randomly sampled negative class examples. For each 512 bp region from the whole genome, the final prediction was made by averaging the predictions of the models that used data not seen during training.

### Model performance

To estimate the model performance, we computed precision, recall, F1, and ROC AUC on the test set (Table 1). For benchmark models, we applied DEEPZ and gradient-boosting methods. DEEPZ model was run with the set of 1,054 omics features as described in Beknazarov et al, 2020 for humans on the Shin et al dataset (Shin et al, 2016). Predictions for the test set and whole genome were done the same way as for Z-DNABERT models. CatBoost (Dorogush et al, 2018 *Preprint*) was selected as a gradient-boosting benchmark model because CatBoost can use categorical features as an input. The boosting model was trained on the same training set as Z-DNABERT and DEEPZ. Each segment from the training set has been encoded into boosting records. Each nucleotide was transformed into a DNA segment with 256 + 5 nucleotides. The DNA segment was decomposed in 256 6-mers, and every 6-mer from this DNA segment was mapped to a number from a set of all possible enumerated 6-mers. The resulting categorical vector of length 256 was subsequently used as an input for a boosting model. The Z-DNA was located in the center of the 256 bp DNA targets. All encoded sequences formed a training set that was randomly down sampled to 400,000 objects because of calculation limitations. Test set measurements were performed on the whole test set encoded in the same way.

### Attention visualization

Attention visualization was done with DNABERT-viz tool as described in the original DNABERT article (Ji et al, 2021).

### BZ junction motif detection

We took coordinates of starts (5′-ends in the plus orientation) and ends (3′-ends also in plus orientation) of Z-DNABERT predictions and extended them by 5 bp upstream and downstream. The resulting 11-bp long sequences were used as input of the MEME motif discovery tool from MEME Suit launched with default settings (Bailey et al, 2015).

### Z-DNA maps

We used existing data available from kethoxal-assisted sequencing studies (Weng et al, 2020; Wu et al, 2020; Wu et al, 2022) and downloaded from GEO (GSE202044, GSE192822, GSE139420). The

Z-HUNT3 program is available at https://github.com/Ho-Lab-Colostate/zhunt.

### Z-DNABERT mutagenesis maps

To produce mutagenesis maps, Z-DNABERT was first run using the original sequence, then for each site, every nucleotide was replaced with the three alternative nucleotides and the effect of each substitution was calculated as the sum of log(1+p) over each sequence position where p is the probability of Z-DNA formation predicted by the model. By adding 1 to p, we avoided problems with taking the log of a zero probability. The approach allows us to take into account the effects of adjacent sequences on Z-DNA formation, incorporating information of junction formation and cooperativity effects that drive the transition. The heatmap shows the effect of each substitution relative to the original sequence, with the ratio of the two scores reflecting the probability that each will form Z-DNA in that particular context.

### Z-flipons overlap with quantitative trait loci and sites of alternative RNA processing

GWAS catalogue data files were downloaded from https://www.ebi.ac.uk/gwas/ (v. 1.0) (Buniello et al, 2019). Data on eQTL, sQTL, and edQTL were downloaded from the GTEx portal https://www.gtexportal.org/ (v 8.0). The Swiss Bioinformatics Institute track for alternative splicing (Iseli et al, 2002) was accessed through the UCSC browser. Annotation for ENCODE cCREs combined from all cell types was downloaded from the UCSC genome browser (data last updated 2020-05-20). Deleterious protein variants were downloaded from the gnomAD-pLOF database (v 2.1.1) (Karczewski et al, 2020).

### Z-flipons overlap with RNA-editing databases

Z-RNA-editing sites from 1,413 genes in lung adenocarcinoma tumors were taken from Sharpnack et al research (Sharpnack et al, 2018). 113 ADAR1 p150-dependent sites were taken from Sun et al, 2021. Editing sites, associated with Alzheimer's disease, were downloaded from the ADeditome database (Wu et al, 2021) and also from Rediportal (http://srv00.recas.ba.infn.it/atlas/search.html) (Lo Giudice et al, 2020).

### RNA structural analysis

RNA secondary structure was predicted with RNA fold from Vienna Package (Hofacker, 2009).

### Haplotype analysis

Haplotypes were determined using the LDLink tool (Myers et al, 2020). Each haplotype was scored by assigning +1 to the alleles that increased trait values and −1 otherwise. For SNPs where quantitative trait measures were unavailable, each allele was assigned a value of 0.

## Data Availability

The code is freely available at: https://github.com/mitiau/Z-DNABERT. We have provided a readme file for the resource at: https://github.com/mitiau/Z-DNABERT/blob/main/README.md. The Z-DNA-BERT tool is freely available at: https://colab.research.google.com/github/mitiau/Z-DNABERT/blob/main/ZDNA-prediction.ipynb. A user can input a sequence of interest into our pretrained model to identify Z-flipons with a high level of confidence.

## Supplementary Information

## Acknowledgements

The work was supported by the grant for research centers in the field of AI provided by the Analytical Center for the Government of the Russian Federation (ACRF) in accordance with the agreement on the provision of subsidies (identifier of the agreement 000000D730321P5Q0002) and the agreement with HSE University No. 70-2021-00139. A Herbert was supported by grants from InsideOutBio, Inc.

### Author Contributions

D Umerenkov: conceptualization, resources, software, formal analysis, validation, investigation, visualization, methodology, and writing—review and editing.
A Herbert: conceptualization, resources, software, formal analysis, supervision, funding acquisition, validation, investigation, visualization, methodology, project administration, and writing—original draft, review, and editing.
D Konovalov: resources, data curation, software, validation, investigation, methodology, and writing—review and editing.
A Danilova: resources, data curation, software, formal analysis, validation, investigation, and writing—review and editing.
N Beknazarov: resources, software, formal analysis, validation, investigation, methodology, and writing—review and editing.
V Kokh: conceptualization, supervision, and funding acquisition.
A Fedorov: software and formal analysis.
M Poptsova: conceptualization, resources, formal analysis, supervision, funding acquisition, validation, investigation, visualization, methodology, project administration, and writing—original draft, review, and editing.

### Conflict of Interest Statement

A Herbert is the founder of InsideOutBio, a company that works in the field of immuno-oncology. The authors declare that the research was conducted in the absence of any commercial or financial relationships that could be construed as a potential conflict of interest.

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
