## [Reviewer comments · Life Science Alliance]

Life Science Alliance

Biological roles for Z-DNA and Z-RNA revealed by Deep Learning

Dmitry Umerenkov, Alan Herbert, Dmitrii Konovalov, Anna Danilova, Nazar Beknazarov, Vladimir Kokh, Aleksandr Fedorov, and Maria Poptsova

DOI: <https://doi.org/N/A>

Corresponding author(s): Alan Herbert, InsideOutBio, Inc

Review Timeline:

Submission Date:	2023-01-31
Editorial Decision:	2023-03-27
Revision Received:	2023-04-13
Editorial Decision:	2023-04-18
Revision Received:	2023-04-24
Editorial Decision:	2023-04-25
Revision Received:	2023-04-25
Accepted:	2023-04-28

Scientific Editor: Novella Guidi

Transaction Report:

March 27, 2023

Re: Life Science Alliance manuscript #LSA-2023-01962-T

Alan Herbert
InsideOutBio, Inc

Dear Dr. Herbert,

Thank you for submitting your manuscript entitled "Biological roles for Z-DNA and Z-RNA revealed by Deep Learning" to Life Science Alliance. The manuscript was assessed by expert reviewers, whose comments are appended to this letter. We invite you to submit a revised manuscript addressing the Reviewer comments.

Thank you for this interesting contribution to Life Science Alliance. We are looking forward to receiving your revised manuscript.

Sincerely,

B. MANUSCRIPT ORGANIZATION AND FORMATTING:

Reviewer #1 (Comments to the Authors (Required)):

Propensity of certain sequence motifs to form non-canonical nucleic acid structures if under torsional strain (best summarized under the term "flipons" by corresponding author Alan Herbert in his corresponding papers), is typically investigated through combinations of statistical, thermodynamical, and structural parameters. Typical and widely accepted software applications to search for the B- to Z-DNA flipon are Z-Catcher and Z-Hunt, though several others were developed for the purpose and available through GitHub.

However, as with other sequence motif-based genome screening algorithms, e.g., transcription factor binding sites, splice sites, wet lab data not always confirm in silico data, where "not always" may approach insignificance biologically and statistically. Of the paper's current authors, Maria Poptsova and Nazar Beknazarov, published a paper in 2020 on their DeepZ algorithm (Beknazarov N, Jin S, Poptsova M. 2020. Deep learning approach for predicting functional Z- DNA regions using omics data. Sci Rep 10) that laid the foundations of the current manuscript. DeepZ is based on convolutional and recurrent neural networks. Now they applied the pretrained transformer DNABERT by Ji et al (Ji Y, Zhou Z, Liu H, Davuluri RV. 2021. DNABERT: pre-trained Bidirectional Encoder Representations from Transformers model for DNA-language in genome, Bioinformatics 37: 2112-2120), resulting in Z-DNABERT. Z-DNABERT significantly outperforms DeepZ with a recall of 0.89, precision of 0.78, and ROC AUC of 0.99. "Z-flipons are enriched in promoters and telomeres and overlap quantitative trait loci for RNA expression, RNA editing, splicing and disease associated variants. The evidence ... was cross-validated across a number of orthogonal databases and provides a curated set of hypotheses for experimental validation" (cited from their manuscript).

Surprisingly - not only to the authors - many of the effects seen are mediated not through Z-DNA but through Z-RNA flipons in transcripts but also in non-coding RNA.

All the main points of the paper are very well supported. Not only by correlative (statistical) evidence, but by quasi causal evidence enabled by an exclusive property of the transformer, i.e., computational mutagenesis to access the effects of a base substitution on Z-DNA formation. The training sets are all well curated, experimentally confirmed databases of molecular biological and cell biological functions. Moreover, aside from its scientific validity, the paper sets standards in the quality and beauty of its graphics and data representations - my honest congratulations on this aspect of the manuscript.

One very minor point that needs amendment: all % numbers in Venn diagrams and distributions graphs need definitions in figure legends (as we tell our students, don't we...?)

Reviewer #2 (Comments to the Authors (Required)):

The manuscript by Umerenkov et al. provides a series of genome-wide analyses that aims to show that DNA/RNA sequences with the potential to form a Z structure (a left-handed double helical structure) are pervasive genomic elements essential for the many genome transactions. The authors call those sequences Z-flipons. It was known for some time that these sequences are enriched in regulatory regions of the genome including promoters. Consequently, structures formed at Z-flipons (especially in the DNA background) were the focus of extensive in vitro and in vivo studies over the last decades. However, their biological roles, the physiological conditions for their formation, and even their existence in vivo are still debated.

Neither existing genome-wide assays for the detection of Z-DNA (ssDNA-Seq, Kas-seq, ChIP-seq) nor the computational method for their detection from the sequence are completely reliable. Both are likely to pick up GC-rich sequences and single-stranded DNA that occur often in mammalian CpG islands, and in promoters. There is no "golden standard" set of promoters that certainly contain Z-DNA with known function, so false negative and positive rates of detection, both experimental and computational, are unknown and at present unknowable.

Based on existed DNABERT model, the authors developed an algorithm for predicting Z-flipons (mostly DNA sequences and to some extent RNA sequences) genome-wide. Importantly, this approach - ZDNABERT - allows to make computational mutagenesis and access to the effects of a base substitution on Z-DNA formation. The authors then cross-correlate predicted Z-flipons with the astonishing number of orthogonal datasets genome-wide and conclude that Z-flipons and their variants are important for a variety of genomic processes and result in different phenotypic outputs.

The research question asked in the manuscript is important. There are several promising results. I think this work lends support to the notion that Z-flipons could play regulatory roles. However, I have two main reservations, detailed below.

MAJOR POINTS:

(1) The authors claim that ZDNABERT algorithm was tuned on the experimental map of Z-DNAs at nucleotide resolution in human cells (Kouzine et al, 2017). In this paper, combined permanganate treatment in vivo and nuclease digest in vitro

introduced DNA breaks at the DNA regions that were in ssDNA conformation at the time of the drug treatment. These breaks are labeled with biotin. Then DNA is sonicated, streptavidin-selected, and sequenced. Thus, all non-B DNA structures (Z-DNA included) are mapped at the same resolution as in Shin et al. 2016. The resolution is determined by the size of DNA fragments after the sonication - usually around 200 bp. To differentiate between structures, Kouzine et al overlapped the resulting signal with computationally predicted sequences potentially able to fold into non-B DNA conformation. Z-DNA motifs (Z-flipons in the terminology of the current manuscript) were predicted using the Z-Hunt program (Champ et al., 2004).

Consequently, (1) I do not see the validity of the statement that Z-DNABERT is based on nucleotide-resolution experimental data; (2) the training model of Z-DNABERT is based on the computational prediction of Z-Hunt, not experimental data alone; (3) a considerable portion of Z-DNA detected by Kouzine et al, might be false positive due to overlap of ssDNA signal with multiple non-B DNA conformations or regions of transcriptional activities. This is especially true for the regions near the transcriptional start site known for high enrichment of ssDNA. Is this the reason that the authors of the current manuscript observed "the maximum overlap of experimental Z-DNA vs predicted (95.32%) is observed in 5' exons <300 bp from the TSS" exactly where high ssDNA signal from the paused RNA polymerase II is expected? The authors should explain what exactly they use for the training, and what are the possible caveats and limitations. In the current form, the first opinion of a reader is that computer training was based on computer-generated predictions, not on experimental data. Is this a reason for the enhancement of Z-DNABERT over DEEPZ? What is the performance of Z-DNABERT in comparison to Z-Hunt?

To make the presented story solid, the authors might try to test Z-DNABERT on orthogonal experimental approaches to detect ssDNA. The Kas-seq approach also maps ssDNA ("KAS-seq: genome-wide sequencing of single-stranded DNA by N3-kethoxal-assisted labeling"). It will be informative to see if Z-DNABERT can catch Z-DNA formation in Kas-seq data and if the yield of Z-DNAs is similar between Kouzine et al data and Kas-Seq. Kouzine et al also map Z-DNA in genome supercoiled in vitro. In this set of data, there is no overlapping between Z-DNA and transcriptional activity. Could authors detect "the maximum overlap of experimental Z-DNA vs predicted (95.32%) is observed in 5' exons <300 bp from the TSS" in this data set?

(2) All over the manuscript there are highly speculative statements. A few examples are below:

Here the authors tried to explain the analysis of Z-RNA stem/loop motif in eQTLs for SMAD1: "The strength of Z-RNA formation associated with each exon likely affects the expression of each isoform. The transcription of isoforms containing exon 3 may be favored by the rs13144151 A allele that disrupts Z-RNA formation and allows RNA polymerase progression. In contrast, both exons 2 and 4 contain strong Z-RNA folds that could cause RNA polymerases, leading to the lower readout of these isoforms."

Do authors believe that Z-RNA formation (or even Z-DNA) may somehow change the progression of RNA polymerase? Is there any evidence or even expectation/models for that?

While the data and analysis presented in the "An eQTL in SCARF2 affects MED15 and Height" section are interesting, it will be purely speculative to conclude that Z-DNA formation is responsible for "increased coupling between enhancer and promoter". The 1 kb resolution of the Hi-C assay is simply not enough to make any kind of link to SNP (rs874100). Other hypotheses are not considered. The negative and positive effect of SNPs on chromatin topology is unknown, at least to this reviewer.

The same goes with multiple references to chromatin topology all over the manuscript:

"The hypotheses generated...establish a close connection between Z-flipons, CTCF and loop formation".

"With Z-DNA, the chromatin structures and condensates formed can enable approximation of distant regions through loop formation independently of loop extrusion (Fudenberg et al. 2016) and capable of maintaining nuclear structure in the absence of cohesin (Schwarzer et al. 2017)."

I do not see any substantial evidence for that. A possible mechanism of "a close connection" was not even proposed.

The manuscript should be extensively rewritten. All not supported speculations as shown here and elsewhere should be removed.

MINOR POINTS:

1) I believe that the author should acknowledge the first study on the detection of Z-DNA genome-wide: "Human genomic Z-DNA segments probed by the Z alpha domain of ADAR1" from Droge lab.

2) It is not clear what "KEx" stands for.

3) Consortium et al. 2020 - I think it is the wrong format of citation.

4) To make sense of Figure 1, d,e (and similar figures elsewhere) the differential enrichment of flipons should be shown together with the uniform signal distribution.

5) Interestingly, we did not detect substantial overlap with regions of G-banding or with high recombination, negating a number

of previous proposals made without experimental support (Supplemental Figure 3)". Supplemental Figure 4?

6)I think that the section "Z-flipons in Action" should belong to the Discussion. Discussion should be more focused.

7)What is the meaning of the red bar in Figure 3, e?

8)Should be explained what is the microC map. I think that for clarity it is better to call it the Hi-C map.

9)The sentence "We were able to define four haplotypes that incorporate other neighborhood SNPs that are also associated with height" is not clear. Are those SNPs in the flippons (experimental? Predicted?)

10)Figure 4 - Not clear if Rs1264670 and RS 12646702 are the same.

11)"The failure of Z-DNABERT to detect many of the Z-RNA in this fold may reflect that the experimental determination of Z-DNA was performed in a single cell line." Not clear. Kouzine et al data was generated for Mouse and Human cells.

12)"(STXBP5L) (Bhatnagar et al. 2011) is short and heavily edited (Figure 6)." Should be supplemental Figure 6.

13)"The novel combinations of ZNF produced by alternative splicing could prevent the escape of recently recombined transposons and viruses from KRAB mediated suppression." - Citation? Explanation?

14)The term "Z-flipon" should be removed from the abstract. Currently, it is used only by the authors of the current manuscript. Its use in the first place will confuse the readers.

15)Many sentences and even paragraphs are written in such a way that it is difficult to understand their real meaning. Here is just one example: "We identified a different motif in which an inverted HR formed a Z-RNA stem by base pairing with another HR. The motif was present in ZNF587B RNA, which has 13 C2H2 (two cysteines, two histidines) ZNF and related proteins that also contain ZNFs and a KRAB domain that suppresses the expression of transposons and viruses by binding to relatively conserved sequences in their genomes. Together this family of proteins constitutes an intracellular form of immunity to protect against such threats (Ecco et al. 2017). Here we provide evidence that the system is adaptive. The HR in these proteins links together adjacent ZNFs (Luchi 2001). The sequence has some remarkable properties." It is written like proteins might have HR that has "purine-pyrimidine inverted repeats capable of forming short Z-prone dsRNA helices"?

16)The manuscript requires serious grammar-editing.

17)The format of different Figures should be unified. Many labels are almost impossible to read.

Reviewer #3 (Comments to the Authors (Required)):

Summary:

A genome-wide approach is used to understand the biology of Z-DNA and RNA. The analysis was completed using DNABERT (transformer-based NLP model that is trained to perform language tasks related to DNA and protein sequences). The study greatly extends an understanding in the language of Z-flipons. This study is of interest as these regulatory elements found in regions of DNA, consisting of long runs of CG dinucleotides, such as in promoter regions of certain genes have shown disease correlation.

The study focuses only on genomic variants that have been reported in GWAS. This leaves scope to find novel variants which could go unnoticed due to the limitations of GWAS.

Methods:

Applied the fine-tune module of DNABERT and rigorously evaluated Z-flipon biology.

Results:

The computational analysis using Z-DNABERT revealed that. Z-flipons were colocalized with cCREs overlapping richly with ends of the chromosomal regions. This is of biological interest as these regions are rich in repetitive DNA sequences, which are prone to genomic rearrangements that can cause various genetic disorders.

The overlaps with various databases focusing on different biological context points in the direction of establishing the important of Z-flipons in health and disease. The examples help in understanding the biology better (eQTL in 5'UTR of SMAD1 gene - associated with HDL cholesterol).

Overall evaluation: The paper in a method application paper to a real-world problem. It discusses the biology and establishes its relation to mendelian diseases, genomic elements and in gene regulation. The GitHub and example colab notebooks are informative and useful. It would be helpful to the biology community to have access to those Z-flipon identified sequences to test their laboratory experiments with relevance to advancing the understanding of these elements.

Recommendation:

Overall, the paper has the potential to make an important contribution to genomics.

The figure 3 H is not clear (blurred). Figure 5D and F need to be enlarged.

Font type and size are inconsistent across the paper. Z-flipons role in regulatory pathways can be explored.

Though the article is nicely written, it would be great if the authors can share the finetuned model for better reusability via GitHub. Adding a readme file will also help the researchers in the future use of the codebase.

Thanks to all three reviewers for their helpful comments and suggestions for improving the manuscript.

We have responded to the issues raised, highlighting the questions asked in yellow with our replies are in line with blue and italicized text.

Overall, we have added comparisons of Z-DNABERT with Z-HUNT3 and Kethoxal assisted sequencing (KAS-seq or as abbreviated in the paper as Kseq). There are now two new figures, one new supplemental figure, three new supplemental tables and 2 new supplemental data sets. Each new item is now discussed in detail in the manuscript, greatly improving the paper.

Reviewer #1 (Comments to the Authors (Required)):

Propensity of certain sequence motifs to form non-canonical nucleic acid structures if under torsional strain (best summarized under the term "flipons" by corresponding author Alan Herbert in his corresponding papers), is typically investigated through combinations of statistical, thermodynamical, and structural parameters. Typical and widely accepted software applications to search for the B- to Z-DNA flipon are Z-Catcher and Z-Hunt, though several others were developed for the purpose and available through GitHub.

However, as with other sequence motif-based genome screening algorithms, e.g., transcription factor binding sites, splice sites, wet lab data not always confirm in silico data, where "not always" may approach insignificance biologically and statistically. Of the paper's current authors, Maria Poptsova and Nazar Beknazarov, published a paper in 2020 on their DeepZ algorithm (Beknazarov N, Jin S, Poptsova M. 2020. Deep learning approach for predicting functional Z- DNA regions using omics data. Sci Rep 10) that laid the foundations of the current manuscript. DeepZ is based on convolutional and recurrent neural networks.

Now they applied the pretrained transformer DNABERT by Ji et al (Ji Y, Zhou Z, Liu H, Davuluri RV. 2021. DNABERT: pre-trained Bidirectional Encoder Representations from Transformers model for DNA-language in genome, Bioinformatics 37: 2112-2120), resulting in Z-DNABERT. Z-DNABERT significantly outperforms DeepZ with a recall of 0.89, precision of 0.78, and ROC AUC of 0.99. "Z-flipons are enriched in promoters and telomeres and overlap quantitative trait loci for RNA expression, RNA editing, splicing and disease associated variants. The evidence ... was cross-validated across a number of orthogonal databases and provides a curated set of hypotheses for experimental validation" (cited from their manuscript).

Surprisingly - not only to the authors - many of the effects seen are mediated not through Z-DNA but through Z-RNA flipons in transcripts but also in non-coding RNA.

All the main points of the paper are very well supported. Not only by correlative (statistical) evidence, but by quasi causal evidence enabled by an exclusive property of the transformer, i.e., computational mutagenesis to access the effects of a base substitution on Z-DNA formation. The training sets are all well curated, experimentally confirmed databases of molecular biological and cell biological functions. Moreover, aside from its scientific validity, the paper sets standards in the quality and beauty of its graphics and data representations - my honest congratulations on this aspect of the manuscript.

One very minor point that needs amendment: all % numbers in Venn diagrams and distributions graphs need definitions in figure legends (as we tell our students, don't we...?)

Thanks for the great review! We have fixed the issues with abbreviations you point out

Reviewer #2 (Comments to the Authors (Required)):

Thank-you for spending so much time on the manuscript and for your very thorough review! We have addressed the major points directly below and by additions or edits to the manuscript. The feedback is really helpful. We also appreciate you drawing our attention to a number of typos that we missed and also your suggestions for improving the figures.

The manuscript by Umerenkov et al. provides a series of genome-wide analyses that aims to show that DNA/RNA sequences with the potential to form a Z structure (a left-handed double helical structure) are pervasive genomic elements essential for the many genome transactions. The authors call those sequences Z-flipons. It was known for some time that these sequences are enriched in regulatory regions of the genome including promoters. Consequently, structures formed at Z-flipons (especially in the DNA background) were the focus of extensive in vitro and in vivo studies over the last decades. However, their biological roles, the physiological conditions for their formation, and even their existence in vivo are still debated.

Neither existing genome-wide assays for the detection of Z-DNA (ssDNA-Seq, Kas-seq, ChIP-seq) nor the computational method for their detection from the sequence are completely reliable. Both are likely to pick up GC-rich sequences and single-stranded DNA that occur often in mammalian CpG islands, and in promoters. There is no "golden standard" set of promoters that certainly contain Z-DNA with known function, so false negative and positive rates of detection, both experimental and computational, are unknown and at present unknowable.

Based on existed DNABERT model, the authors developed an algorithm for predicting Z-flipons (mostly DNA sequences and to some extent RNA sequences) genome-wide. Importantly, this approach - ZDNABERT - allows to make computational mutagenesis and access to the effects of a base substitution on Z-DNA formation. The authors then cross-correlate predicted Z-flipons with the astonishing number of orthogonal datasets genome-wide and conclude that Z-flipons and their variants are important for a variety of genomic processes and result in different phenotypic outputs.

The research question asked in the manuscript is important. There are several promising results. I think this work lends support to the notion that Z-flipons could play regulatory roles. However, I have two main reservations, detailed below.

MAJOR POINTS:

(1) The authors claim that ZDNABERT algorithm was tuned on the experimental map of Z-DNAs at nucleotide resolution in human cells (Kouzine et al, 2017). In this paper, combined permanganate treatment in vivo and nuclease digest in vitro introduced DNA breaks at the DNA regions that were in ssDNA conformation at the time of the drug treatment. These breaks are labeled with biotin. Then DNA is sonicated, streptavidin-selected, and sequenced. Thus, all non-B DNA structures (Z-DNA included) are mapped at the same resolution as in Shin et al. 2016. The resolution is determined by the size of DNA fragments after the sonication - usually around 200 bp. To differentiate between structures, Kouzine et al overlapped the resulting signal with

computationally predicted sequences potentially able to fold into non-B DNA conformation. Z-DNA motifs (Z-flipons in the terminology of the current manuscript) were predicted using the Z-Hunt program (Champ et al., 2004).

Consequently, (1) I do not see the validity of the statement that Z-DNABERT is based on nucleotide-resolution experimental data; (

The Kouzine data relies on mapping unpaired thymines present in the two B-Z junctions formed with B-DNA at either end of a Z-DNA helix. It tests for features associated with alternative DNA conformations that are specified prior to experiment. The prediction of the conformation formed is based on the presence of distinct patterns of single-strandedness should a particular sequence form an alternative structure. In the case of Z-DNA, the confirmation depends on the detection of two B-Z junctions that specifies the sequence that forms Z-DNA at nucleotide resolution. The approach is analogous to the way that CLIP=seq is used to map protein binding sites. The reviewer is correct that it is not possible to sometimes deconvolute a particular experimental result into a single possible alternative structure as noted by Kouzine et al. in their paper where they state that “3%, 11%, 12%, and 18% of Z-DNA, G4, SIDD, and H-DNA, respectively, overlap with other non-B DNA structures over 50% of their length”.

To address these points, we have added this paragraph to the introduction

“Our approach starts with the nucleotide resolution, permanganate/S1 nuclease dataset (KEx) from Kouzine et al. that is based on mapping unpaired thymines present in the two B-Z junctions formed with B-DNA at either end of a Z-DNA helix (Kouzine F et al, 2017). Following training of Z-DNABERT with this training set, we compared the predictions with those from orthogonal approaches based on Z-HUNT3 (Ho PS, 2009) and Kethoxal- Assisted Sequencing (Kseq). Z-HUNT3 is based on in vitro measurements capturing the energetic cost of flipping a basepair from B-DNA to Z-DNA, using a fixed energy cost for the formation of two B-Z junctions. It estimates the propensity of a sequence to form Z-DNA in supercoiled DNA. Kseq uses chemical modification of unpaired guanosine bases with azide-tagged kethoxal performed with intact cells. The reaction detects regions of single-stranded DNA arising from active transcription and R-loop formation (Weng X et al, 2020, Wu T et al, 2020). Unlike $KMnO_4$ that detects the unpaired base at a B-Z junction, Kseq captures the opening of a Z-forming sequence as it flips from one conformation to another. The half-life for the B-Z transition in vitro under physiological levels of supercoiling is estimated to be 100 ms. Consequently, the reaction of kethoxal with DNA is measured over 5 – 10 minutes compared to the 70s used for $KMnO_4$ modification and Kseq does not provide specific information about junctions (Jovin TM et al, 1987). Here, we use the overlap between each of these different predictive and experimental approaches to map Z-DNA formation to specific genomic loci. We confirm the enrichment of Z-flipons in promoters and identify Z-DNA prone repeat families.”

2) the training model of Z-DNABERT is based on the computational prediction of Z-Hunt, not experimental data alone;

Actually Kouzine et al do not use Z-HUNT. They use the NCBI non-B-DNA prediction tool that is based on sequence motifs, not energetics of Z-DNA formation ([3](https://nonb-T variant (XP_016884554.1:p.Thr425Ser), which is not an eQTL but rather a sQTL and the intron 6 variant rs882745 (NM_153334.7:c.1203-97G>T) that is just upstream of an alternative splice site for SCARF2 (Figure 3H). While many of SNPs do not overlap flippons, they help to define haplotypes associated with high and low expression of MED15."

10) Figure 4 - Not clear if Rs1264670 and RS 12646702 are the same.

Thanks for catching this mistake - fixed

11) "The failure of Z-DNABERT to detect many of the Z-RNA in this fold may reflect that the experimental determination of Z-DNA was performed in a single cell line." Not clear. Kouzine et al data was generated for Mouse and Human cells.

Here we are only analyzing human data for which only a single cell line was used

12) "(STXBP5L) (Bhatnagar et al. 2011) is short and heavily edited (Figure 6)." Should be supplemental Figure 6.

Thanks - corrected

13) "The novel combinations of ZNF produced by alternative splicing could prevent the escape of recently recombined transposons and viruses from KRAB mediated suppression." - Citation? Explanation?

14)The term "Z-flipon" should be removed from the abstract. Currently, it is used only by the authors of the current manuscript. Its use in the first place will confuse the readers.

The term flipon is used by authors besides up in their papers, but not usually as a key word for PubMed Indexing. Its use has been accepted by a number of editors at different Journals, including Nature and the Royal Society. The word flipon captures a concept that has been well explained in terms of repats that are prone to form alternative conformation, their frequency in the genome, their natural selection and their genetics. The concept is easy to explain to non-scientists. On the other hand, there are still a few older scientists who resist the use of the term prion to describe how proteins can induce transmissible disease, but that viewpoint is no longer as widely held view as it was initially. Similar to prions, flipons are tricky to work with, both from the scientific and societal perspective.

15)Many sentences and even paragraphs are written in such a way that it is difficult to understand their real meaning. Here is just one example: "We identified a different motif in which an inverted HR formed a Z-RNA stem by base pairing with another HR. The motif was present in ZNF587B RNA, which has 13 C2H2 (two cysteines, two histidines) ZNF and related proteins that also contain ZNFs and a KRAB domain that suppresses the expression of transposons and viruses by binding to relatively conserved sequences in their genomes. Together this family of proteins constitutes an intracellular form of immunity to protect against such threats (Ecco et al. 2017). Here we provide evidence that the system is adaptive. The HR in these proteins links together adjacent ZNFs (Iuchi 2001). The sequence has some remarkable properties." It is written like proteins might have HR that has "purine-pyrimidine inverted repeats capable of forming short Z-prone dsRNA helices"?

Agreed and these sections and others have been rewritten

16)The manuscript requires serious gramma-editing.

We have done our best with the new revision.

17)The format of different Figures should be unified. Many labels are almost impossible to read.

We noted formatting issues that arose from converting between .svg and raster images. We have addressed this problem and enlarged many panels where the type was previously on the smaller side.

Reviewer #3 (Comments to the Authors (Required)):

Thank-you taking time to review the manuscript and for your helpful suggestions. We really appreciate your comments.

Summary:

A genome-wide approach is used to understand the biology of Z-DNA and RNA. The analysis was completed using DNABERT (transformer-based NLP model that is trained to perform language tasks related to DNA and protein sequences). The study greatly extends an understanding in the language of Z-flipons. This study is of interest as these regulatory elements found in regions of DNA, consisting of long runs of CG dinucleotides, such as in promoter regions of certain genes have shown disease correlation. The study focuses only on genomic variants that have been reported in GWAS. This leaves scope to find novel variants which could go unnoticed due to the limitations of GWAS.

Methods:

Applied the fine-tune module of DNABERT and rigorously evaluated Z-flipon biology.

Results:

The computational analysis using Z-DNABERT revealed that. Z-flipons were colocalized with cCREs overlapping richly with ends of the chromosomal regions. This is of biological interest as these regions are rich in repetitive DNA sequences, which are prone to genomic rearrangements that can cause various genetic disorders.

The overlaps with various databases focusing on different biological context points in the direction of establishing the important of Z-flipons in health and disease. The examples help in understanding the biology better (eQTL in 5'UTR of SMAD1 gene - associated with HDL cholesterol).

Overall evaluation: The paper in a method application paper to a real-world problem. It discusses the biology and establishes its relation to mendelian diseases, genomic elements and in gene regulation. The GitHub and example colab notebooks are informative and useful. It would be helpful to the biology community to have access to those Z-flipon identified sequences to test their laboratory experiments with relevance to advancing the understanding of these elements.

Recommendation:

Overall, the paper has the potential to make an important contribution to genomics.

The figure 3 H is not clear (blurred). Figure 5D and F need to be enlarged.

Thanks for pointing this out – we have followed your recommendations and enlarged these figures.

Font type and size are inconsistent across the paper. Z-flipons role in regulatory pathways can be explored.

Though the article is nicely written, it would be great if the authors can share the finetuned

model for better reusability via GitHub. Adding a readme file will also help the researchers in the future use of the codebase.

Great suggestion – we have added a readme file along with a standalone notebook for the prediction of Z-DNA in a sequence of interest,

<https://github.com/mitiau/Z-DNABERT/blob/main/README.md>

April 18, 2023

Re: Life Science Alliance manuscript #LSA-2023-01962-TR

Dr. Alan Herbert
InsideOutBio, Inc
42 8th Street
Charlestown 02129-4221

Dear Dr. Herbert,

Thank you for submitting your revised manuscript entitled "Biological roles for Z-DNA and Z-RNA revealed by Deep Learning" to Life Science Alliance. The manuscript has been seen by one original reviewer whose comments are appended below. While the reviewer continue to be overall positive about the work in terms of its suitability for Life Science Alliance, some important issues remain.

Our general policy is that papers are considered through only one revision cycle; however, given that the suggested changes are relatively minor, we are open to one additional short round of revision. Please note that I will expect to make a final decision without additional reviewer input upon resubmission.

Please submit the final revision within one month, along with a letter that includes a point by point response to the remaining reviewer comments.

To upload the revised version of your manuscript, please log in to your account: <https://lsa.msubmit.net/cgi-bin/main.plex>
You will be guided to complete the submission of your revised manuscript and to fill in all necessary information.

B. MANUSCRIPT ORGANIZATION AND FORMATTING:

Sincerely,

Reviewer #2 (Comments to the Authors (Required)):

The authors have revised their manuscript and addressed a few comments raised in my first review. However, my major point #1 and associated questions about the data set the authors used for the training of Z-DNABERT were totally not resolved. For clarity, I repeat my comment below:

FROM THE FIRST REVIEW: "(1) The authors claim that ZDNABERT algorithm was tuned on the experimental map of Z-DNAs at nucleotide resolution in human cells (Kouzine et al, 2017). In this paper, combined permanganate treatment in vivo and nuclease digest in vitro introduced DNA breaks at the DNA regions that were in ssDNA conformation at the time of the drug treatment. These breaks are labeled with biotin. Then DNA is sonicated, streptavidin-selected, and sequenced. Thus, all non-B DNA structures (Z-DNA included) are mapped at the same resolution as in the Shin et al. 2016. The resolution is determined by the size of DNA fragments after the sonication - usually around 200 bp. To differentiate between structures, Kouzine et al overlapped the resulting signal with computationally predicted sequences potentially able to fold into non-B DNA conformation. Z-DNA motifs (Z-flipons in terminology of the current manuscript) were predicted using the Z-Hunt program (Champ et al., 2004).

Consequently, (1) I do not see the validity of the statement that Z-DNABERT is based on nucleotide-resolution experimental data; (2) training model of Z-DNABERT is based on the computational prediction of Z-Hunt, not experimental data alone; (3) considerable portion of Z-DNA detected by Kouzine et al, might be false positive due to overlap of ssDNA signal with multiple non-B DNA conformations or regions of transcriptional activities. This is especially true for the regions near transcriptional start site known for high enrichment of ssDNA. Is this the reason that the authors of current manuscript observed "the maximum overlap of experimental Z-DNA vs predicted (95.32%) is observed in 5' exons <300 bp from the TSS" exactly where high ssDNA signal from the paused RNA polymerase II is expected?

The authors should explain what exactly they use for the training, and what are the possible caveats and limitations. In the current form, the first opinion of reader is that computer training was based on computer generated predictions, not on experimental data. Is this a reason for the enhancement of Z-DNABERT over DEEPZ? What is the performance of Z-DNABERT in comparison to Z-Hunt?

To make the presented story solid, the authors might try to test Z-DNABERT on orthogonal experimental approaches to detect ssDNA. Kas-seq approach also map ssDNA (KAS-seq: genome-wide sequencing of single-stranded DNA by N3-kethoxal-assisted labeling). It will be informative to see if Z-DNABERT can catch Z-DNA formation in Kas-seq data and if the yield of Z-DNAs is similar between Kouzine et al data and Kas-Seq.

Kouzine et al also map Z-DNA in genome supercoiled in vitro. In this set of data there is no overlapping between Z-DNA and transcriptional activity. Could authors detect "the maximum overlap of experimental Z-DNA vs predicted (95.32%) is observed in 5' exons <300 bp from the TSS" in this data set?"

From what I see in the edited manuscript and the rebuttal, the authors still believe that the experimental set they used for the training specifies the sequence that forms Z-DNA at nucleotide resolution. As written in the rebuttal - "The Kouzine data relies on mapping unpaired thymines present in the two B-Z junctions formed with B-DNA at either end of a Z-DNA helix. It tests for features associated with alternative DNA conformations that are specified prior to experiment. The prediction of the conformation formed is based on the presence of distinct patterns of single-strandedness should a particular sequence form an alternative structure. In the case of Z-DNA, the confirmation depends on the detection of two B-Z junctions that specifies the sequence that forms Z-DNA at nucleotide resolution."

It is an incorrect interpretation of Kouzine et al data. Basically, Kouzine et al computationally predicted the sequences with Z-DNA forming potential and then looked at the enrichment of ssDNA in the 500 bp window centered at these sequences.

There is a clear description of this procedure in that paper: "For each occurrence of SMnB, we counted the number of sequencing tags in two windows: the signal window of 500 bp length centered at the motif midpoint, and the local background window of 2×250 bp length whose two parts are located on both sides of the signal window and are adjacent to it. We computed a p value of observed number of tags in signal window assuming the total number of tags in both windows using binomial distribution and taking into account tags' mappability in these windows. If the mappability in the signal windows was too low (below 20%) the motif was removed from the analysis. To test whether the ssDNA signal at SMnB was higher than expected by chance we used a permutation test. However, SMnB are not uniformly distributed across a genome. To control for this, we computed the total number of unique tags in all signal and background windows across the genome. We then randomly distributed the same number of tags in these windows and computed the number of randomized tags in signal and local background windows for each SMnB and the corresponding p values. Based on real and randomized data we computed p value thresholds for a range of false discovery rate (FDR) values."

The authors of the current manuscript claimed that " Kouzine et al do not use Z-HUNT. They use the NCBI non-B-DNA prediction tool that is based on sequence motifs, not energetics of Z-DNA formation (<https://nonb-4abcc.ncifcrf.gov/apps/site/default>). Neither of these tools incorporates sequence information about B-Z forming sequences, although Z-HUNT does give the same energetic penalty for all."

It is mistaken again. As indicated in Kouzine et al "Occurrences of sequence motifs of non-B DNA (SMnB) were identified in silico in the mouse (mm9) and human (hg19) genomes. Potential Z-DNA motifs were predicted using the Z-Hunt online server with default settings (Champ et al., 2004)."

With these slipups in the first place, the authors were not able to answer any further questions associated with the major point.

IN SUMMARY: The authors should explain clearly what exactly they use for the training, and what are the possible caveats and limitations.

Responses to Reviewers

Thanks for the passing on the additional comments of reviewer 2 and allowing us to address them. We have added two new figures (Figure 2C and Supplemental Figure 3E) and more text to the address specific points.

In response to concerns over the use of “nucleotide resolution” in reference to the Kouzine dataset we used.

1. We removed “nucleotide resolution” when we made any direct reference to the Kouzine method. Specifically, we deleted from the main text
 - a. Our approach starts with ~~the nucleotide resolution~~, permanganate/S1 nuclease dataset (KEx) from Kouzine et al. that is based on mapping unpaired thymines present in the two B-Z junctions formed with B-DNA at either end of a computationally predicted Z-DNA helix (Kouzine F et al, 2017).
 - b. experiments with a resolution of 100-150 basepairs (bp) (Shin SI et al, 2016) and the ~~nucleotide resolution~~, permanganate/S1 nuclease dataset (KEx) from Kouzine et al. that is based on mapping unpaired thymines present in the two B-Z junctions formed with B-DNA at either end of a
 - c. While Z-DNABERT is based on ~~nucleotide resolution~~ sequence data, DEEPZ analyzes DNA fragments
 - d. With Z-DNABERT trained on the KEx ~~nucleotide resolution~~ experimental data, we generated genome-wide whole genome maps of Z-DNA prone regions (Supplemental Data 2),
 - e. Z-flipons by tuning the transformer algorithm implemented in DNABERT (Ji Y et al, 2021) with experimentally ~~validated derived~~ Z-DNA forming sequences obtained from the human genome ~~at nucleotide resolution through KMnO₄ mapping~~.
2. From the Supplemental Table 2 legend, we edited “The model based on the ~~nucleotide resolution~~ experimental Kouzine et al” to replace “nucleotide resolution” with “experimental.”
3. We made additional changes to the text to address specific points raised by reviewer 2.
 - a. We expanded the first paragraph of the results to better differentiate between the KEx data and the single-stranded DNA seq on which that summary is based. Our changes are highlighted in yellow.

“Currently there are two human experimental datasets available that provide information on Z-DNA formation within human cells: the Shin et al ChIP-seq (Chromatin Immunoprecipitation followed by DNA sequencing of fragments) experiments with a resolution of 100-150 basepairs (bp) (Shin SI et al, 2016) and the experimentally based dataset from Kouzine et al. (KEx) (Kouzine F et al, 2017). ~~Kouzine et al determined the regions of Z-DNA formation the overlap of~~

unpaired thymines detected using permanganate/S1 nuclease sequencing (ssDNA-seq) with Z-DNA forming sequences predicted by Z-HUNT3. The thymines subject to modification were used to define the two B-Z junctions formed with B-DNA at either end of a Z-DNA helix. The approach employed a number of statistical corrections to identify ssDNA-seq signals solely due to RNA polymerase 2 transcription or to other sequence variations (Kouzine F et al, 2017). The final set (KEx) with all non-B-DNA (NoB) structures annotated is referred to by the authors as “ssDNA + SMnB”. Both Shin et al. and Kouzine et al. approaches were performed in intact cells and differed from an earlier approach where Z α was diffused into detergent permeabilized cells and then cross-linked to DNA using formaldehyde over a number of hours (Li H et al, 2009).

- b. We added a sentence to the following paragraph on page 8 to further clarify the difference. Here we explicitly describe how Kouzine et al. use Z-HUNT as requested by the reviewer who was correct in pointing out our mistake in our previous response where we incorrectly referred to the non-B-DNA database that defines a Z-DNA motif as “G followed by Y (C or T) for at least 10 nt; One strand must be alternating Gs”. Our changes are highlighted in yellow.

“Also displayed are the binding sites for AGO1 and AGO2, proteins guided by microRNA seed sequence matches with proximal promoter nucleotides (Herbert A et al, 2023). All the sequencing methods reveal an increase in unpaired bases at promoters. The KEx approach adjusts for ssDNA formed in the absence of NoB structures by both probabilistic approaches based on randomizing counts in a region and by calculating expected counts after excluding SINE repeat sequences from their analysis. They also used thresholds to identify regions where the ssDNA-seq counts are twofold higher than expected from RNA Polymerase 2 transcription (Kouzine F et al, 2017). The regions with excess ssDNA hits were overlapped with predictions of Z-HUNT3 to define the regions of Z-DNA formation shown here in the KEx track. No filtering was performed for Kseq results. Indeed, the strand specific sequencing (plus or minus tracks) shown for HeLa cells were used for the detection of R-loops formed when an RNA:DNA hybrid displaces an unpaired DNA strand from a double-stranded DNA helix (Wu T et al, 2022). The results indicate that the Z-DNA predictions for both Z-HUNT3 and Z-DNABERT align with only a small fraction of the ssDNA detected by Kseq, with both strands undergoing modification.

The dotted boxes in Figure 2 highlight the different patterns of overlap between the approaches we examined. Panel a shows concordance of Z-DNABERT mappings by all approaches...”

4. We also performed additional analysis to bolster our conclusions where we looked for motifs in the B-Z junctions predicted by Z-DNABERT and present the results in Figure 2C and in Supplemental figure 3E. We added this section to page 9. Our changes are highlighted in yellow.

“Our Z-DNABERT results enabled us to whether certain motifs are enriched in B-Z junctions, as junctional sequences are not used by Z-HUNT3 to predict Z-DNA formation and the KEx training set is 7 times smaller in genomic coverage than the predicted set. We found that a d(TAAA) motif was enriched in the 5' region at both ends of the Z-DNABERT junction between B-DNA and Z-DNA (Figure 2C). The result is consistent with in vitro studies showing that adenosines from B-Z junctions well (Ha SC et al, 2005, Kim D et al, 2018) and differ from those found using just the KEx dataset (Supplemental Figure 3E). Further our finding supports the suggestion that some sequences do not support B-Z junction formation. Instead, these sequences oppose the flip to Z-DNA by an otherwise Z-prone sequence (Kim D et al, 2018).”

5. We also addressed the concerns of reviewer 2 further in the discussion. Our changes are highlighted in yellow .

“The Z-DNA forming regions were detected experimentally through reagents such as KMnO₄ and kethoxal that detect unpaired bases. the enrichment we find in promoters occurs in regions where the high levels of negative supercoiling detected by other means (Georgakopoulos-Soares I et al, 2022, Kouzine F et al, 2013, Teves SS & Henikoff S, 2013) are sufficient to induce a flip from B-DNA to Z-DNA. The KEX approach was designed to partition the ssDNA regions detected into those associated with RNA polymerase transcription and those in which NoB forming sequences were associated with higher-than-expected KMnO₄ modification. This method is validated by the strand-specific Kseq results presented in Figure 2 that were designed to detect R-loops formed during active transcription. In contrast to R-loops, Z-DNA formation produces modifications either to both DNA strands in a region or is not associated with their formation. Our findings also suggest that certain motifs favor the formation of B-Z junctions. Consistent with in vitro studies, we detected a strong preference for adenosines at a B-Z junction (Ha SC et al, 2005, Kim D et al, 2018). The result stems directly from the Z-DNABERT algorithm as Z-HUNT3 is agnostic to B-Z junction sequence and assigns the same penalty to all. Yet, we saw a genome-wide enrichment of adenosines at B-Z junctions with the exclusion of other bases. The strongest motif found by Z-DNABERT (d(TAAA) at the 5' end of the junction (Figure 2C) was not apparent in the KEx dataset (Supplemental Figure 3E), even though ~15% of segments were common to both sets. , This type of B-Z junction is likely favored at the ends of the d(AC)_n repeats that are also enriched in this motif.

The repeat adenosines in a B-Z junction will likely affect the local DNA conformation. In crystal structures, the B-Z junction has an 11° bend that could be extended by the additional adenosines we find present in the in vivo data, due to a narrowing of the B-DNA minor groove in short adenosine repeats (Hizver J et al, 2001). The conformational flexibility of a B-Z junction that results from the additional adenosines may also facilitate intercalation by small molecules. The extra adenosines will form B-DNA, with the minor groove representing a preferred docking site for high mobility group proteins (Bewley CA et al, 1998,

Strahs D & Schlick T, 2000). The biological effects of this class of B-Z junctions requires additional investigation.

6. We expanded on the methods to describe the data we used from the Kouzine et al. paper and added a section to describe the motif analysis for B-Z junctions. Our changes are highlighted in yellow.

a. **Experimental Z-DNA training data**

Permanganate/S1 Nuclease Footprinting Z-DNA data contained 41 324 regions with total length of 773 788 bp in human. We downloaded the data set "ssDNA + SMnB" from

<https://www.ncbi.nlm.nih.gov/CBBresearch/Przytycka/index.cgi#nonbdna> (Kouzine F et al, 2017). We verified the mapping to hg19 and filtered out ENCODE blacklisted regions. The ssDNA wig file was downloaded from the same location. For DNABERT the data was preprocessed by converting a sequence into 6-mer representation.

b. **B-Z junction motif detection.**

We took coordinates of starts (5' ends in the plus orientation) and ends (3' ends also in plus orientation) of ZDNABERT predictions and extended them by 5 bp upstream and downstream. Resulting 11 bp long sequences were used as input of MEME motif discovery tool from MEME Suit launched with default settings (Bailey TL et al, 2015).

April 25, 2023

RE: Life Science Alliance Manuscript #LSA-2023-01962-TRR

Dr. Alan Herbert
InsideOutBio, Inc
42 8th Street
Charlestown 02129-4221

Dear Dr. Herbert,

Thank you for submitting your revised manuscript entitled "Biological roles for Z-DNA and Z-RNA revealed by Deep Learning". We would be happy to publish your paper in Life Science Alliance pending final revisions necessary to meet our formatting guidelines.

- please upload both your main and supplementary files as single files and add a separate figure legend section to your main manuscript text
- please make sure that the author order in the manuscript text and the order in our system match and that every author listed in the manuscript is entered in our system

To upload the final version of your manuscript, please log in to your account: <https://lsa.msubmit.net/cgi-bin/main.plex>. You will be guided to complete the submission of your revised manuscript and to fill in all necessary information. Please get in touch in case you do not know or remember your login name.

A. FINAL FILES:

B. MANUSCRIPT ORGANIZATION AND FORMATTING:

Sincerely,

April 28, 2023

RE: Life Science Alliance Manuscript #LSA-2023-01962-TRRR

Dr. Alan Herbert
InsideOutBio, Inc
42 8th Street
Charlestown 02129-4221

Dear Dr. Herbert,

Thank you for submitting your Research Article entitled "Biological roles for Z-DNA and Z-RNA revealed by Deep Learning". It is a pleasure to let you know that your manuscript is now accepted for publication in Life Science Alliance. Congratulations on this interesting work.

DISTRIBUTION OF MATERIALS:

Again, congratulations on a very nice paper. I hope you found the review process to be constructive and are pleased with how the manuscript was handled editorially. We look forward to future exciting submissions from your lab.

Sincerely,
